# Training Recurrent Neural Networks Online by Learning Explicit State Variables

**Somjit Nath, Vincent Liu, Alan Chan, Xin Li, Adam White and Martha White**
Department of Computing Science
University of Alberta
`{somjit,vliu1,achan4,xzli,amw8,whitem}@ualberta.ca`

## Abstract

Recurrent neural networks (RNNs) allow an agent to construct a state-representation from a stream of experience, which is essential in partially observable problems. However, there are two primary issues one must overcome when training an RNN: the sensitivity of the learning algorithm's performance to truncation length and and long training times. There are variety of strategies to improve training in RNNs, the mostly notably Backprop Through Time (BPTT) and by Real-Time Recurrent Learning. These strategies, however, are typically computationally expensive and focus computation on computing gradients back in time. In this work, we reformulate the RNN training objective to explicitly learn state vectors; this breaks the dependence across time and so avoids the need to estimate gradients far back in time. We show that for a fixed buffer of data, our algorithm—called Fixed Point Propagation (FPP)—is sound: it converges to a stationary point of the new objective. We investigate the empirical performance of our online FPP algorithm, particularly in terms of computation compared to truncated BPTT with varying truncation levels.

## 1 Introduction

Many online prediction problems are partially observable: the most recent observation is typically insufficient to make accurate predictions about the future. Augmenting the inputs with a history can improve accuracy, but can require a long history when there are long-term dependencies back in time. Recurrent Neural Networks (RNNs) (Elman, 1990; Hopfield, 1982) learn a state which *summarizes* this history. Specifically, RNNs contain recurrent connections to their hidden layers which allow past information to propagate through time. This state need not correspond to a true underlying state; rather, the state is subjective and constructed to facilitate prediction. RNNs have been widely used, in speech recognition (Hinton *et al.*, 2012; Graves *et al.*, 2013; Miao *et al.*, 2015; Chan *et al.*, 2016), image captioning (Mao *et al.*, 2014; Lu *et al.*, 2016; Vinyals *et al.*, 2014), speech synthesis (Mehri *et al.*, 2016) and reinforcement learning (Hochreiter and Schmidhuber, 1997; Düll *et al.*, 2012).

Despite these success, there are significant stability and computational issues in training RNNs online (Pascanu *et al.*, 2013; Tallec and Ollivier, 2017). In the *online* setting, the agent faces an unending stream of data and on each step the agent must update its parameters to make a new prediction. RNNs are typically trained either using Backpropagation-through-time (BPTT) (Werbos, 1990) or approximations to an algorithm called Real-Time Recurrent Learning (RTRL) (Williams and Zipser, 1989a; Pearlmutter, 1995), although there are methods that appeal to other principles (see Murray (2019) for instance). The update for BPTT is a variant of standard backpropagation, computing gradients all the way back in time. This approach is problematic because the computational cost scales linearly with the number of time-steps. A more common alternative is truncated BPTT (T-BPTT) (Williams and Peng, 1990) which only computes the gradient up to some maximum number of steps: we truncate how far back in time we unroll the network to update the parameters. This approximation, though, is not robust to long-term dependencies (Tallec and Ollivier, 2017). Approximate gradients can also be computed online by RTRL (Williams and Zipser, 1989b). This online algorithm, however, has high computational complexity per step and therefore is not commonly used in practice.

Recently, there have been some efforts towards approximating gradients for back-propagation, both for feedforward NNs and RNNs. Synthetic gradients and BP($\lambda$) (Jaderberg *et al.*, 2017) use an idea similar to returns from reinforcement learning: they approximate gradients by bootstrapping off estimated gradients in later layers (Jaderberg *et al.*, 2017; Czarnecki *et al.*, 2017). There are several methods that approximate RTRL—which is itself an approximation of the true gradient back in time—including NoBackTrack (Ollivier and Charpiat, 2015), Unbiased Online Recurrent Optimization (UORO) (Tallec and Ollivier, 2017) which uses an unbiased rank-1 approximation to the full matrix gradient, and Kronecker Factored RTRL (Mujika *et al.*, 2018) which uses a Kronecker product decomposition to approximate the RTRL update for a class of RNNs. Finally, there are some methods that use selective memory back in time to compute gradients for the most pertinent samples, using skip connections (Ke *et al.*, 2017). All of these methods, however, attempt to approximate the gradient back in time, for the current observation and state, and so suffer to some extent from the same issues as BPTT and RTRL.

In this paper, we investigate an alternative optimization strategy that does not attempt to approximate the gradient back in time. Instead we learn the state variables in the RNN explicitly. These new variables are optimized to both improve prediction accuracy, and to maintain consistency in producing the next learned state variables. This second constraint is a fixed-point formula for the states under the given RNN dynamics.[1] We develop a provably sound stochastic update for the new fixed-point objective, which we then use to develop an online algorithm for training RNNs. The algorithm explicitly optimizes state vectors and RNN parameters with many efficient one-step—or short term multi-step updates—across a buffer. Instead of focusing computation to get a more accurate gradient estimates for this time-step, our algorithm, called Fixed Point Propagation (FPP), can more effectively use computation to update prediction accuracy across states. We demonstrate that the algorithm is effective on several problems with long-term dependencies, and improves over T-BPTT, particularly in terms of stability and computation.

## 2 PROBLEM SETTING AND BACKGROUND

We consider a partially observable online setting, where an immediate observation is not sufficient for prediction. More formally, assume there is a sequence of $n$ observations, $\mathbf{o}_1, \ldots, \mathbf{o}_n$, which provide only partial information about an unknown underlying sequence of states. After obtaining an observation $\mathbf{o}_i$, the agent makes a prediction $\hat{\mathbf{y}}_i$ and sees the actual outcome $\mathbf{y}_i$. The goal is to minimize this prediction error. Given only $\mathbf{o}_i$, the agent is unlikely to make accurate predictions about $\mathbf{y}_i$, because $\mathbf{o}_i$ is not a sufficient statistic to predict $\mathbf{y}_i$: $p(\mathbf{y}|\mathbf{o}_i, \mathbf{o}_{i-1}, \mathbf{o}_{i-2}, \ldots) \neq p(\mathbf{y}|\mathbf{o}_i)$. The agent could obtain a lower prediction error using a history of observations. Unfortunately, the agent may require a prohibitively long history, even if this history could have been summarized compactly.

An alternative is to construct state using a Recurrent Neural Network (RNN), by learning a state-update function. Given the current (constructed) state $\mathbf{s}_{t-1} \in \mathbb{R}^k$, and a new observation $\mathbf{o}_t \in \mathbb{R}^d$, the parameterized state-update function $f_{\mathbf{W}} : \mathbb{R}^k \times \mathbb{R}^d \to \mathbb{R}^k$, with parameters $\mathbf{W}$, produces the next (constructed) state $\mathbf{s}_t = f_{\mathbf{W}}(\mathbf{s}_{t-1}, \mathbf{o}_t)$. For example, $f_{\mathbf{W}}$ could be a linear weighting of $\mathbf{s}_{t-1}$ and $\mathbf{o}_t$, with a ReLu activation: $f_{\mathbf{W}}(\mathbf{s}_{t-1}, \mathbf{o}_t) = \max([\mathbf{s}_{t-1}, \mathbf{o}_t]\mathbf{W}, \mathbf{0})$. More complex state-updates are possible, like the gating in Long Short-Term Memory (LSTM) (Hochreiter and Schmidhuber, 1997).

The objective is to adapt these parameters $\mathbf{W}$ for the state-update to minimize prediction error. For the current state $\mathbf{s}_t$, a prediction is made by parameterized function $g_{\boldsymbol{\beta}} : \mathbb{R}^k \to \mathbb{R}^m$ with learned parameters $\boldsymbol{\beta}$. For example, the prediction could be a linear weighting of the state, $g_{\boldsymbol{\beta}}(\mathbf{s}) = \boldsymbol{\beta}^\top \mathbf{s}$. We denote the prediction error as $\ell_{\boldsymbol{\beta}} : \mathbb{R}^k \times \mathbb{R}^m \to \mathbb{R}$ for a given $\boldsymbol{\beta}$. For example, this loss could be

$$\ell_{\boldsymbol{\beta}}(\mathbf{s}_t; \mathbf{y}_t) = \|g_{\boldsymbol{\beta}}(\mathbf{s}_t) - \mathbf{y}_t\|_2^2.$$

---

[1]Recurrent Backpropagation and related variants (Almeida, 1987; Pineda, 1987; Scellier and Bengio, 2017; Liao *et al.*, 2018) also use fixed points for their optimization, but in a different way. These algorithms only address a restricted class of RNNs, that assume a fixed input and converge to a single low-energy state—a fixed point of the dynamics for that given input. These RNNs are actually highly related to Graph NNs (Scarselli *et al.*, 2008), because the temporal nature only arises from cyclic connections, rather than from temporal data. Their problem setting is fundamentally different from our online prediction setting, and is usually used for associative memory with Hopfield networks or semi-supervised problems. Recurrent Backpropagation cannot be used for our setting and so we do not further discuss this class of RNN algorithms.

The goal in RNNs training is to minimize, for some start state $\mathbf{s}_0$,

$$\min_{\boldsymbol{\beta}, \mathbf{W}} \sum_{i=1}^{n} \ell_{\boldsymbol{\beta}}(f_{\mathbf{W}}(...f_{\mathbf{W}}(\underbrace{f_{\mathbf{W}}(\mathbf{s}_0, \mathbf{o}_1)}_{\mathbf{s}_1}, \mathbf{o}_2), ..., \mathbf{o}_i); \mathbf{y}_i). \tag{1}$$

Computing gradients for this objective, however, can be prohibitively expensive. A large literature on optimizing RNNs focuses on approximating this gradient, either through approximations to RTRL or improvements to BPTT. RTRL (Williams and Zipser, 1989b) uses a recursive gradient form, which can take advantage of gradients computed up until the last observation to compute the gradient for the next observation. This estimate, however, is only exact in the offline case and thus RTRL is an approximation of the true gradient in our online setting. Further, in either online or offline, RTRL requires $O(k^4)$ computation per observation (recall $k$ is the dimension of the state). In BPTT, gradients are computed back in time, by unrolling the network. In the online setting, it is infeasible to compute gradients all the way back to the beginning of time. Instead, this procedure is truncated $T$ steps back in time. T-BPTT is suitable for the online setting, and requires $O(Tk^2)$ computation per step, i.e., for each observation.

Arguably the most widely-used strategy is T-BPTT, because of its simplicity. Unfortunately, T-BPTT has been shown to fail in settings where dependencies back in time are further than $T$ (Tallec and Ollivier, 2017), as we affirm in our experiments. Yet, the need for simple algorithms remains. In this work, we investigate an alternative direction for optimizing RNNs that does not attempt to estimate the gradients of (1).

Note that in addition to a variety of optimization strategies, different architectures have also been proposed to facilitate learning long-term dependencies with RNNs. The most commonly used are LSTMs (Hochreiter and Schmidhuber, 1997), which use gates to remember and forget components of the state. Other architectures include clockwork RNNs (Koutník et al., 2014), phased LSTMs (Neil et al., 2016), hierarchical multi-scale RNNs (Chung et al., 2016), dilated RNNs (Chang et al., 2017), and skip RNNs (Campos et al., 2017). In this work, we focus on a general purpose RNN algorithm, that could be combined with each of these architectures for further improvement.

## 3 A NEW FIXED-POINT OBJECTIVE FOR RNNs

In this section we introduce our new formulation for training RNNs. We begin with an idealized setting to introduce and explain the ideas. Later we will generalize our approach to partially observable online training tasks.

First, assume the observations are produced by an underly Markov Chain with a discrete set of states, and the agent has access to a set of observations that are deterministic function of the state. We denote the set of states $\mathcal{H} = \{1, \dots, n\}$, and the observations from each state as $\mathbf{o}_1, \dots, \mathbf{o}_n$. The goal is to find state vectors $\mathbf{s}_1, \dots, \mathbf{s}_n \in \mathbb{R}^k$ that satisfy two goals. One is to enable the state to be updated

$$f_{\mathbf{W}}(\mathbf{s}_i, \mathbf{o}_j) = \mathbf{s}_j \quad \forall j \text{ where } \mathbf{P}(i, j) > 0 \tag{2}$$

for $\mathbf{P} : \mathcal{H} \times \mathcal{H} \to [0, 1]$ the transition dynamics. Another criterion is for these state vectors to facilitate accurate predictions. In particular, the learned state should minimize $\ell_{\boldsymbol{\beta}}(\mathbf{s}_j; \mathbf{y}_j)$ for all $h$, where $\mathbf{y}_j \in \mathbb{R}$ is the expected target for a true state $j$. Together, this results in the following optimization, with the relationship between states encoded as a constraint

$$\min_{\boldsymbol{\beta}, \mathbf{W}, \mathbf{s}} \sum_{i,j \in \mathcal{H}} \mathbf{P}(i, j) \ell_{\boldsymbol{\beta}}(f_{\mathbf{W}}(\mathbf{s}_i; \mathbf{o}_j), \mathbf{y}_j) \qquad \text{s.t. } f_{\mathbf{W}}(\mathbf{s}_i, \mathbf{o}_j) = \mathbf{s}_j \quad \forall i, j \text{ where } \mathbf{P}(i, j) > 0$$

The satisfiability of this will depend on $f_{\mathbf{W}}$ and if $\mathbf{s}_i$ and $\mathbf{o}_j$ can uniquely determine $\mathbf{s}_j$.

More generally, we will not know the underlying state, nor is it necessarily discrete. But, we can consider a similar objective for observed data. Assume $n$ observations have been observed, $\mathbf{o}_1, \dots, \mathbf{o}_n$, with corresponding targets $\mathbf{y}_1, \dots, \mathbf{y}_n$. Let the state variables be stacked in a matrix $\mathbf{S} \in \mathbb{R}^{k \times n}$ and observations as a matrix $\mathbf{O} \in \mathbb{R}^{d \times n}$, with $\mathbf{S} = [\mathbf{s}_0, \dots, \mathbf{s}_n]$ and $\mathbf{O} = [\mathbf{o}_1, \dots, \mathbf{o}_n]$. The constraint on the states becomes $\mathbf{S} = F_{\mathbf{W}}(\mathbf{S}, \mathbf{O})$ for operator

$$F_{\mathbf{W}}(\mathbf{S}, \mathbf{O}) \stackrel{\text{def}}{=} [\mathbf{S}_{:,0}, f_{\mathbf{W}}(\mathbf{S}_{:,0}, \mathbf{O}_{:,1}), ..., f_{\mathbf{W}}(\mathbf{S}_{:,n-1}, \mathbf{O}_{:,n})]. \tag{3}$$

We call this the fixed-point constraint, since a solution $\mathbf{S}$ to the constraint is a fixed point of the system defined by $F_{\mathbf{W}}(\cdot, \mathbf{O})$. The resulting optimization, for this batch, is

$$\min_{\boldsymbol{\beta}, \mathbf{W}, \mathbf{S}} \sum_{i=1}^{n} \ell_{\boldsymbol{\beta}}(f_{\mathbf{W}}(\mathbf{s}_{i-1}, \mathbf{o}_i); \mathbf{y}_i) \quad \text{s.t. } \mathbf{S} = F_{\mathbf{W}}(\mathbf{S}, \mathbf{O}). \tag{4}$$

The solution to this new optimization corresponds to the solution for the original RNN problem in (1)—when also optimizing over $\mathbf{s}_0$ in (1)—because the fixed-point constraint forces variables $\mathbf{s}_i$ to be representable by $f_{\mathbf{W}}$. Therefore, the reformulation as a fixed point problem has not changed the solution; rather, it has only made explicit that the goal is to learn these states and facilitates the use of alternative optimization strategies.

Reformulations like the one in (4) have been widely considered in optimization, because (4) is actually an auxiliary variable reformulation of (1). In this case, the auxiliary variables are the states $\mathbf{S}$. Using auxiliary variables is a standard strategy in optimization—under the general term *method of multipliers*—to decouple terms in an optimization and so facilitate decentralized optimization.

Such auxiliary variable methods have even been previously considered for optimizing neural networks. Carreira-Perpiñán and Wang (2014) introduced the Method of Auxiliary Coordinates (MAC), which explicitly optimize hidden vectors in the neural network. Taylor *et al.* (2016) proposed a similar strategy, but introduced an additional set of auxiliary variables to obtain further decoupling and a particularly efficient algorithm for the batch setting. Scellier and Bengio (2017) introduced Equilibrium Propagation for symmetric neural networks, where the state of the network is explicitly optimized to obtain a stationary point in terms of the energy function. Gotmare *et al.* (2018) built on these previous ideas to obtain a stochastic gradient descent update for distributed updates to blocks of weights in a neural network. Our proposed optimization can be seen as a variation of the objective considered for MAC (Carreira-Perpiñán and Wang, 2014, Equation 1), though we arrived at it from a different perspective: with the goal to learn explicit state vectors.

The objective in (4) still has two issues. First, it is not amenable to online updating: it is a batch optimization with a constraint. Second, it does not allow for any training back in time. But, this stringent computational restriction is unnecessary. We could have instead asked: learn states so that when iterated twice through the RNN, the resulting state enables accurate predictions on the target two steps in the future. We develop a more general objective below to address both issues.

We can rewrite the objective so that it is clear how to stochastically sample it, and so enable online updating. As in MAC-QP (Carreira-Perpiñán and Wang, 2014), we reformulate this constrained objective into an unconstrained objective with a quadratic penalty, with $\lambda > 0$

$$L(\boldsymbol{\beta}, \mathbf{W}, \mathbf{S}) \stackrel{\text{def}}{=} \frac{1}{n} \sum_{i=1}^{n} \ell_{\boldsymbol{\beta}}(f_{\mathbf{W}}(\mathbf{s}_{i-1}, \mathbf{o}_i); \mathbf{y}_i) + \frac{\lambda}{2n} \sum_{i=1}^{n} \|\mathbf{s}_i - f_{\mathbf{W}}(\mathbf{s}_{i-1}, \mathbf{o}_i)\|_2^2 \tag{5}$$

Once in this unconstrained form, we can perform stochastic gradient descent on this objective in terms of $\boldsymbol{\beta}, \mathbf{W}$ and $\mathbf{S}$ to reach a stationary point. To use stochastic gradient descent, the objective needs to break up into a sum of losses, $L(\boldsymbol{\beta}, \mathbf{W}, \mathbf{S}) = \frac{1}{n} \sum_{i=1}^{n} L_i(\boldsymbol{\beta}, \mathbf{W}, \mathbf{S})$, where we define

$$L_i(\boldsymbol{\beta}, \mathbf{W}, \mathbf{S}) \stackrel{\text{def}}{=} \ell_{\boldsymbol{\beta}}(f_{\mathbf{W}}(\mathbf{s}_{i-1}, \mathbf{o}_i); \mathbf{y}_i) + \frac{\lambda}{2} \|\mathbf{s}_i - f_{\mathbf{W}}(\mathbf{s}_{i-1}, \mathbf{o}_i)\|_2^2.$$

We can stochastically sample $i$ from our buffer of $n$ samples and update our variables with $\nabla L_i$. Fortunately, because the state variables break connections across time, this gradient is zero for most variables, except $\boldsymbol{\beta}, \mathbf{W}, \mathbf{s}_{i-1}$ and $\mathbf{s}_i$. Therefore, each stochastic update can be computed efficiently.

Second, we can generalize this objective to incorporate more than one step of propagation back in time, simply by generalizing the fixed-point operator. Consider the more general $T$-step fixed point problem $\mathbf{S} = F_{T, \mathbf{W}}(\mathbf{S}, \mathbf{O})$ where

$$F_{T, \mathbf{W}}(\mathbf{S}, \mathbf{O}) \stackrel{\text{def}}{=} \Big[\mathbf{S}_{:,0}, \mathbf{S}_{:,1}, \ldots, \mathbf{S}_{:,T-1}, \underbrace{f_{\mathbf{W}}(\ldots f_{\mathbf{W}}(f_{\mathbf{W}}(\mathbf{S}_{:,0}, \mathbf{O}_{:,1}), \mathbf{O}_{:,2}), \ldots), \mathbf{O}_{:,T})}_{\hat{\mathbf{S}}_{:,T}}, \ldots$$

$$f_{\mathbf{W}}(\ldots f_{\mathbf{W}}(f_{\mathbf{W}}(\mathbf{S}_{:,n-T-1}, \mathbf{O}_{:n-T}), \mathbf{O}_{:,n-T+1}), \ldots), \mathbf{O}_{:,n})\Big].$$

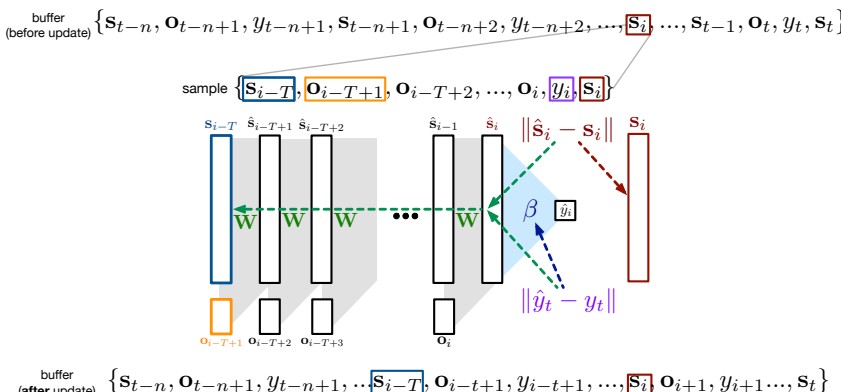

Figure 1: A single update by FPP. It randomly samples $i$, and performs a gradient descent update to $\mathbf{s}_{i-T}, \mathbf{s}_i, \mathbf{W}$ and $\boldsymbol{\beta}$, where the loss on the targets affects $\mathbf{s}_{i-T}, \mathbf{W}, \boldsymbol{\beta}$ and the loss producing the next state variable $\mathbf{s}_i$ affects $\mathbf{s}_{i-T}, \mathbf{s}_i, \mathbf{W}$. The state variables are stored in the buffer, but are explicit variables we learn, just like $\mathbf{W}$ and $\boldsymbol{\beta}$.

For $T = 1$, we recover the operator provided in (3). This generalization mimics the use of $T$-step methods for learning value functions in reinforcement learning. This generalization provides more flexibility in using the allocated computation per step. For example, for a budget of $T$ updates per step, we could use $T$ 1-step updates, $T/2$ 2-step updates, all the way up to one $T$-step update.

The loss for general $T$ similarly decomposes into a sum $\frac{1}{n-T+1} \sum_{i=T}^{n} L_i(\boldsymbol{\beta}, \mathbf{W}, \mathbf{S})$ for

$$L_i(\boldsymbol{\beta}, \mathbf{W}, \mathbf{S}) = \ell_{\boldsymbol{\beta}}(\hat{\mathbf{s}}_i(\mathbf{s}_{i-T}, \mathbf{W}); \mathbf{y}_i) + \frac{\lambda}{2}\|\mathbf{s}_i - \hat{\mathbf{s}}_i(\mathbf{s}_{i-T}, \mathbf{W})\|_2^2 \tag{6}$$

$$\text{where} \quad \hat{\mathbf{s}}_i(\mathbf{s}_{i-T}, \mathbf{W}) \stackrel{\text{def}}{=} f_{\mathbf{W}}(\ldots f_{\mathbf{W}}(f_{\mathbf{W}}(\mathbf{s}_{i-T-1}, \mathbf{o}_{i-T}), \mathbf{o}_{i-T+1}), \ldots), \mathbf{o}_i).$$

For each stochastic sample $i$, $\nabla L_i$ is only non-zero for $\boldsymbol{\beta}, \mathbf{W}, \mathbf{s}_{i-T}$ and $\mathbf{s}_i$. Though these updates can simply be computed using automatic differentiation on $L_i$, the explicit updates are simple so we include them here, using shorthand $\hat{\mathbf{s}}_i$ for $\hat{\mathbf{s}}_i(\mathbf{s}_{i-T}, \mathbf{W})$:

$$\begin{aligned}
\nabla_{\mathbf{s}_{i-T}} L_i &= [\nabla_{\hat{\mathbf{s}}_i} \ell_{\boldsymbol{\beta}}(\hat{\mathbf{s}}_i; \mathbf{y}_i) - \lambda(\mathbf{s}_i - \hat{\mathbf{s}}_i)]^{\top} \nabla_{\mathbf{s}_{i-T}} \hat{\mathbf{s}}_i \\
\nabla_{\mathbf{s}_i} L_i &= \lambda(\mathbf{s}_i - \hat{\mathbf{s}}_i) \\
\nabla_{\mathbf{W}} L_i &= [\nabla_{\hat{\mathbf{s}}_i} \ell_{\boldsymbol{\beta}}(\hat{\mathbf{s}}_i; \mathbf{y}_i) - \lambda(\mathbf{s}_i - \hat{\mathbf{s}}_i)]^{\top} \nabla_{\mathbf{W}} \hat{\mathbf{s}}_i \\
\nabla_{\boldsymbol{\beta}} L_i &= \nabla_{\boldsymbol{\beta}} \ell_{\boldsymbol{\beta}}(\hat{\mathbf{s}}_i; \mathbf{y}_i)
\end{aligned} \tag{7}$$

The online algorithm uses these updates on a sliding window buffer, instead of a fixed buffer. This algorithm—called Fixed Point Propagation (FPP)— is summarized in Figure 1 and Algorithm 1.

As alluded to, the advantage of FPP over T-BPTT is that we are not restricted to focusing all computation to estimate the gradient $T$-steps back in time for one state-observation pair. Rather, instead of sweeping all the way back, we spread value by using updates on random transitions in the buffer. This has three advantages. First, it updates more states per step, including updates towards their targets. Second, this ensures that targets for older transitions are constantly being reinforced, and spends gradient computation resources towards this goal, rather than spending all computation on computing a more exact gradient for the recent time step. This distributes updates better across time, and should likely also result in a more stable state. Third, the formulation as stochastic gradient descent on the fixed point objective makes it a sound strategy—as opposed to truncation which is not sound. FPP, therefore, maintains the simplicity of T-BPTT, but provides a more viable direction to obtain sound algorithms for training RNNs.

## 4    CONVERGENCE RESULTS

In this section we show two theoretical results. First, we show that the FPP algorithm converges to a stationary point, for a fixed buffer. This result is a relatively straightforward application of recent theory for nonconvex optimization (Ghadimi *et al.*, 2016), mainly requiring us to show that our

---

**Algorithm 1** Fixed Point Propagation (FPP)

---

**Input**: a truncation parameter $T$, mini-batch size $B$, and number of updates per step $M$
Initialize weights $\mathbf{W}$ and $\boldsymbol{\beta}$ randomly
Initialize state $\mathbf{s}_0 \leftarrow \mathbf{0} \in \mathbb{R}^d$
Initialize an empty buffer B of size $N$
**for** $t \leftarrow 1, 2, ...$ **do**
    **if** B is full **then**
        Remove the oldest transition
    **end if**
    Observe $\mathbf{o}_t, \mathbf{y}_t$, and compute $\mathbf{s}_t = f_{\mathbf{W}}(\mathbf{s}_{t-1}, \mathbf{o}_t)$
    Add $(\mathbf{s}_t, \mathbf{o}_t, \mathbf{y}_t)$ to buffer B
    **if** $t \geq T$ **then**
        **for** $j \leftarrow 1, \cdots, M$ **do**                      ▷ Multiple updates
            Sample a mini-batch of size $B$, of trajectories of length $T$, from the buffer:
                $\{(\mathbf{s}_{i_l-T}, \mathbf{o}_{i_l-T}, \ldots, \mathbf{s}_{i_l}, \mathbf{o}_{i_l}, \mathbf{y}_{i_l})\}_{l=1}^{B}$ where $i_l$ is the index of the $l$-th mini-batch
            Compute the average mini-batch loss and update $\{\mathbf{s}_{i_l-T}, \mathbf{s}_{i_l}\}_{l=1}^{B}, \mathbf{W}$ and $\boldsymbol{\beta}$
            Update $\{\mathbf{s}_{i_l-T}, \mathbf{s}_{i_l}\}_{l=1}^{B}$ in the buffer
        **end for**
    **end if**
**end for**

---

algorithm can be written as an instance of that framework and to show that each stochastic gradient update is unbiased. This convergence result, nonetheless, is key, as it suggests that FPP is a sound strategy for using replay with RNNs. Previous attempts to use replay for RNNs, in reinforcement learning, were not able to show convergence (Kapturowski *et al.*, 2019), which is to be expected as truncated BPTT updates on a buffer may not be sound.

Additionally, we show that as $\lambda$ approaches infinity, the set of stationary points of the FPP objective approaches the set of stationary points for the RNN objective. In our experiments, we use $\lambda = 1$, as obtaining precisely the same solutions as the RNN objective is not our goal. We include this theoretical result nonetheless for completeness to characterize the relationship between the stationary points of FPP objective and the RNN objective. The proof is similar to that for MAC-QP (Carreira-Perpiñán and Wang, 2014), with the main novelty in checking the KKT conditions for our objective and for linear independence in the Jacobian. Full proofs for both results are in Appendix A.

## 4.1 Convergence of FPP to a Stationary Point for a Fixed Buffer

Recent work uses the idea of randomized gradient descent to show convergence to a stationary point for nonconvex objectives (Ghadimi *et al.*, 2016), as opposed to typical restrictions such as convexity or the PL condition (Karimi *et al.*, 2016). The randomized approach uses a random stopping time $R$, and characterizes the norm of the expected gradient for the variables at this random time. The variables we learn are $(\mathbf{W}, \boldsymbol{\beta}, \mathbf{S}) \in \mathbb{R}^z$, where $z$ is the appropriate dimension.

For the proof we also require the variables to remain in a closed, convex set, to ensure that our objective is Lipschitz. To do so, we will analyze our update with the addition of a projection operator onto a closed ball $C$ in $\mathbb{R}^z$ of radius $r > 0$ about the origin. $r$ can be very large, and we emphasize that $C$ is only a convenience used for theoretical analysis. In practice, we do not project our iterates. Since $\mathbb{R}^d$ is a Hilbert space and $C$ is closed and convex, we have the existence of a unique projection operator $\Gamma$

$$\Gamma(\mathbf{W}_0, \boldsymbol{\beta}_0, \mathbf{S}_0) \stackrel{\text{def}}{=} \underset{(\mathbf{W}, \boldsymbol{\beta}, \mathbf{S}) \in C}{\arg\min} \|(\mathbf{W}_0, \boldsymbol{\beta}_0, \mathbf{S}_0) - (\mathbf{W}, \boldsymbol{\beta}, \mathbf{S})\|^2. \tag{8}$$

Our objective is $L(\mathbf{W}, \boldsymbol{\beta}, \mathbf{S}) \stackrel{\text{def}}{=} \frac{1}{n-T+1} \sum_{i=T}^{n} L_i(\mathbf{W}, \boldsymbol{\beta}, \mathbf{S})$, for $L_i$ defined in Equation (6), for $n > T$ samples. Each time we perform an update, we randomly sample $k_t \sim \text{uniform-}(T, n)$, inclusive of both endpoints. The update to parameters at time $t$, for stepsize $\alpha_t$, is

$$(\mathbf{W}_{t+1}, \boldsymbol{\beta}_{t+1}, \mathbf{S}_{t+1}) \stackrel{\text{def}}{=} \Gamma((\mathbf{W}_t, \boldsymbol{\beta}_t, \mathbf{S}_t) - \alpha_t \nabla L_{k_t}(\mathbf{W}_t, \boldsymbol{\beta}_t, \mathbf{S}_t)). \tag{9}$$

**Theorem 1.** *Let $D$ be a Lipschitz constant of $\nabla L(\mathbf{W}, \boldsymbol{\beta}, \mathbf{S})$. Define probability mass functions*

$$P_N(k) := \frac{\alpha_k - D\alpha_k^2}{\sum_{j=1}^{N} \alpha_j - D\alpha_j^2}.$$

*for each $N \in \mathbb{N}$. Let $R$ be distributed according to $P_N$. Assume $\alpha_t = \frac{1}{2D}$ for all $t$ and that we perform $N$ stochastic updates. Write $x_R = (\mathbf{W}_R, \boldsymbol{\beta}_R, \mathbf{S}_R)$. Then*

$$\mathbb{E}\left[\frac{1}{\alpha_R^2} \|\Gamma(\alpha_R \nabla L(x_R))\|^2\right] = \mathcal{O}\left(\frac{1}{N}\right).$$

## 4.2 RECOVERING RNN SOLUTIONS

Consider the standard RNN problem,

$$\min_{\boldsymbol{\beta}, \mathbf{W}, \mathbf{s}_0} E(\mathbf{W}, \boldsymbol{\beta}, \mathbf{s}_0) \quad \text{for } E(\mathbf{W}, \boldsymbol{\beta}, \mathbf{s}_0) \stackrel{\text{def}}{=} \sum_{i=1}^{n} \ell_{\boldsymbol{\beta}}(f_{\mathbf{W}}(\cdots f_{\mathbf{W}}(f_{\mathbf{W}}(\mathbf{s}_0, \mathbf{o}_1), \mathbf{o}_2), \cdots, \mathbf{o}_i); \mathbf{y}_i) \quad (10)$$

where we also optimize over $s_0$. Our goal is to show that for increasing $\lambda$, the set of stationary points of the FPP objective in Equation (5) approach stationary points of the RNN objective in Equation (10). We assume $T = 1$ in our analysis of FPP.

**Theorem 2.** *Assume we have a positive, increasing sequence $\{\lambda_k\} \to \infty$, a non-negative sequence $\{\epsilon_k\} \to 0$, and a sequence of points $\{(\mathbf{W}_k, \boldsymbol{\beta}_k, \mathbf{S}_k)\}$ such that $\|\nabla L(\mathbf{W}_k, \boldsymbol{\beta}_k, \mathbf{S}_k); \lambda_k)\| \leq \epsilon_k$ for*

$$L(\mathbf{W}_k, \boldsymbol{\beta}_k, \mathbf{S}_k); \lambda_k) \stackrel{\text{def}}{=} \frac{1}{n} \sum_{i=1}^{n} \ell_{\boldsymbol{\beta}}(f_{\mathbf{W}}(\mathbf{s}_{i-1}, \mathbf{o}_i); \mathbf{y}_i) + \frac{\lambda_k}{2} \|\mathbf{s}_i - f_{\mathbf{W}}(\mathbf{s}_{i-1}, \mathbf{o}_i)\|_2^2 \quad (11)$$

*Assume further that $\{(\mathbf{W}_k, \boldsymbol{\beta}_k, \mathbf{S}_k)\}$ has a convergent subsequence $\{(\mathbf{W}_{k_i}, \boldsymbol{\beta}_{k_i}, \mathbf{S}_{k_i})\}$ with limit $(\mathbf{W}^*, \boldsymbol{\beta}^*, \mathbf{S}^*)$. Then $(\mathbf{W}^*, \boldsymbol{\beta}^*, \mathbf{S}^*)$ is a KKT point of the constrained FPP objective (see (12)) and $(\mathbf{W}^*, \boldsymbol{\beta}^*, \mathbf{s}_0^*)$ is a KKT point of the RNN objective (10). Further, if $(\mathbf{W}^*, \boldsymbol{\beta}^*, \mathbf{S}^*)$ is a local min of the constrained FPP objective, then $(\mathbf{W}^*, \boldsymbol{\beta}^*, \mathbf{s}_0^*)$ is a local min of (10).*

## 5 EXPERIMENTAL RESULTS

We designed a sequence of experiments in real and synthetic problems to evaluate our new method compared with several common baselines and to highlight the robustness of our method to different truncation lengths, buffer sizes and number of updates. In particular we compare (1) against T-BPTT with a variety of truncation lengths greater and lesser than the temporal delay required to solve each problem; (2) No-Overlap T-BPTT, a common a variant of T-BPTT that updates on disjoint partitions of the data; and (3) FPP without the state update, which is similar to the *Stored State* T-BPTT algorithm (Kapturowski *et al.*, 2019). We begin by describing the problems we used to evaluate our methods, and why they were chosen. Unless otherwise stated, we report average performance over all training steps (online performance), averaged over 30 independent runs.

**Simulation Problems**   We used two small simulation problems to highlight the robustness of each method to increasing temporal delay in online training. The first tasks is a simple ring of states, called *Cycle World*. On each timestep the agent deterministically transitions to the next state in the chain. The agent's observation is zero in every state, except the last. The agent's objective is to predict the next observation, which is difficult without a memory of length equal to the length of the cycle. With a shorter memory, the agent cannot tell when the last non-zero observation, which is essential for predicting the next observation. This task has been used exclusively in benchmarking k-Markov methods, POMDPs, and predictive state representations (Tanner and Sutton, 2005). The complexity of the task can be easily varied, and yet the determinism ensures the variance does not introduce confounding factors. At each time step, we measure the prediction accuracy for the next observation.

We also experimented with a stochastic prediction task, where correct prediction requires remembering two independent observation's from the past. In particular, the target on the next timestep is probabilistically dependent on the one-dimensional observation 15 timesteps ago and 30 timesteps ago. The dynamics are summarised in Table 1, in Appendix C. This task is called *Stochastic World*.

For this problem, a cross-entropy loss of 0.66 or higher indicates that the learned state did not capture the observation from either 15 or 30 steps in the past. If the state captures the observation from 15 time-steps ago the cross entropy loss is about 0.51. Optimal performance in this problem results in a cross-entropy loss is about 0.46. Like Cycle World, Stochastic World requires a long and detailed memory of past observations, but the stochastic nature of the target pose an additional challenge.

**Real DataSets**   We also performed experiments on two fixed datasets, to gain insights into how each method performed on better known benchmark tasks. In both cases the data was processed and performance evaluated in an online fashion. The first problem is Sequential MNIST. The objective is to classify numerical digits based on a stream of pixel inputs. On each timestep the input is one row (1x28) of the image, and the target is the label of the image. We used an RNN architecture with 512 hidden units as in previous work (Arjovsky *et al.*, 2015). It is not possible to predict the target image base on a few samples, so we wait until 15 steps (corresponding to 15x28 pixels) to begin measuring the error. Here, we report these incorrect predictions for the last 15 time-steps for every image. We ran this on 1000 images, which correspond to 28000 steps.

Finally, we also include results on a a character prediction problem called Penn Tree Bank dataset. This problem is relevant because language modelling remains an important application of recurrent learning systems, and robust performance on this dataset can provide insight into the utility of our new method in application. We used a vocabulary size of 10000. The Target Loss function used here is a weighted cross-entropy loss for a sequence of logits. We used an LSTM with 200 hidden nodes as this architecture was found to perform well in previous work (Zaremba *et al.*, 2014).

**Comparison to T-BPTT**   We compare FPP to T-BPTT for varying truncation levels. For all the algorithms, we used a constant buffer size of 100 and the trajectory length T for both T-BPTT(overlap and no overlap versions) and FPP. All algorithms use $O(T)$ computation per step. For overlap T-BPTT, we employ T-BPTT online by taking each observation and updating with respect to the loss at that time-step, using a T-step truncated gradient. The no-overlap version of T-BPTT performs a batch update for every T observations, such that these observations do not overlap.

Additionally, we include UORO (Tallec and Ollivier, 2017) as another baseline. UORO uses an unbiased rank-1 approximation to approximate RTRL. It is a relatively new method for training RNN online, and has only been tested on small scale datasets. In our experiment, we include memory-1 and rank-1 UORO in Cycleworld and StochasticWorld.

We first compare the performances of FPP and T-BPTT on Cycleworld with varying p. We expect T-BPTT to degrade with $T$ less than the dependence back in time (the length of the cycle $p$); we therefore test both $T = p$ and $T = p/2$ for increasing $p$. To make the results comparable across $p$, we report performance as the ratio to a simple baseline of predicting 0 at every time step. From Figure 2, we can see that FPP is more robust to $T$, whereas T-BPTT with $T = p/2$ performs poorly even when given more data (Figure 2(b)). In early learning, with fewer samples, FPP has an even more clear advantage. Even though T-BPTT can eventually learn optimal predictions for $T = p$, it takes longer than FPP which learns near optimal predictions in early learning (Figure 2(a)).

We additionally compare FPP and the two variants of T-BPTT across all four problems, under different settings of $T$, shown in Figure 3. Across all problems, FPP outperforms the other two for every $T$, except $T = 1$ in CycleWorld where all three methods perform similarly. The performance of FPP is notably better for smaller $T$, as compared to T-BPTT. For example, in Figure 3(b) 20-BPTT has a high loss and is unable to learn both the dependencies, whereas FPP with T=20, performs almost as well as 40-BPTT. Similar conclusions can be made for $T \in \{3, 5\}$ in (a), $T \in \{10, 15, 20, 30\}$ in (b), $T \in \{7, 14, 21, 28\}$ in (c) and $T \in \{1, 5, 10, 20\}$ in (d).

**Benefits of mini-batch updates and multiple updates per step**   One of the advantages of using a buffer is the ability to perform mini-batch updates and multiple updates per step. We evaluate the performance of FPP with and without state updates using M updates per step and a mini-batch of size B. We show the performance with varying $T$. To show the effect of multiple update, we fix $B$ and vary $M \in \{1, 2, 4, 8, 16\}$. To show the effect of mini-batch update, we fix $M$ and vary $B \in \{1, 2, 4, 8, 16\}$. We use a buffer size of 1000 and 10000 training steps.

We also include FPP without state updating, to determine if the benefits of FPP are mainly due to using a buffer rather than due to the new objective to learn explicit state variables. We particularly expect FPP to outperform FPP without state updating under more updates per step, because we showed converge for FPP on a fixed buffer whereas no such result exists for FPP without state

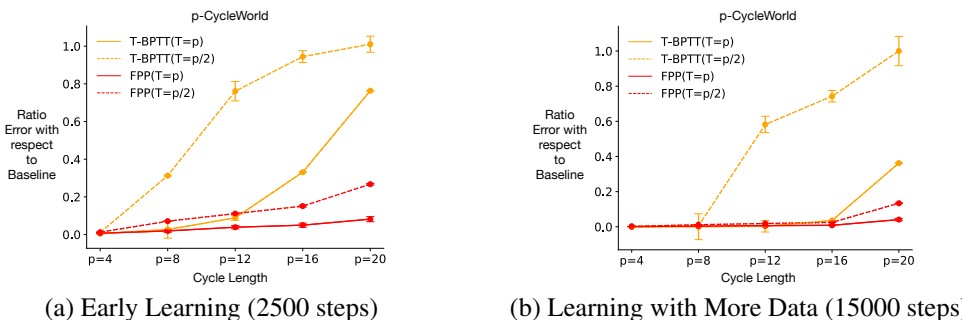

(a) Early Learning (2500 steps)  (b) Learning with More Data (15000 steps)

Figure 2: The ratio error of each of the algorithms with respect to the baseline of predicting 0 at every time step is our measure of performance. For all the values of p, FPP seems to be more robust to T, especially with larger p. The numbers are average over 30 runs with standard error bars.

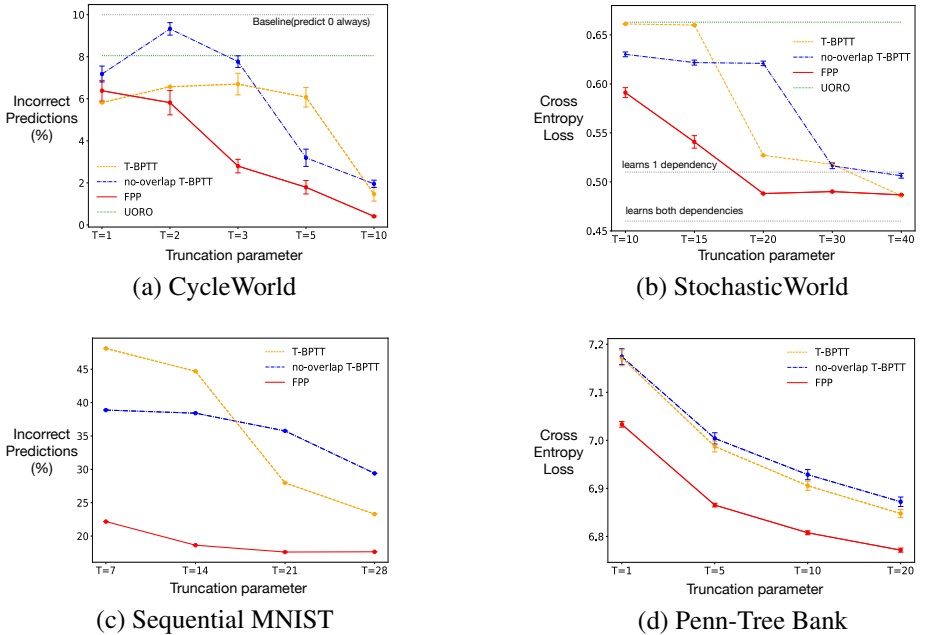

(a) CycleWorld  (b) StochasticWorld

(c) Sequential MNIST  (d) Penn-Tree Bank

Figure 3: Average online performance for FPP (red), T-BPTT (orange) and No-Overlap T-BPTT (blue). Across all the domains, FPP seems to be more robust to T, and it does much better than T-BPTT especially for small T. The numbers are average over 30 runs with one standard error with (a) being run for 5000 steps, (b) for 10000 steps, (c) for 1000 images (28000 steps) and (d) for 5000 steps (5000 points in dataset, processed in order). FPP at $T = 20, 30, 40$ reaches a final solution with optimal performance; it is only above the second line because the plot shows average performance across all steps, rather than final performance.

updating. Here, the buffer is not fixed, but performing more updates per step should move the FPP solution closer to a stationary point of the current buffer.

Figure 4 (a) and (b) shows the effect of multiple updates and (c) and (d) the effect of mini-batch updates. For both, increasing the number of updates and the size of the mini-batch improves performance, except for a bit of overfitting we observed in Stochastic World for increasing updates ($B = 1, M = 16$). However, in general, FPP can better take advantage of both multiple updates and mini-batch updating. The most noticeable gaps are for $T = 16$ and $T = 32$ in StochasticWorld and $T = 1$ and $T = 2$ in CycleWorld. The theory suggests that more updates, even with $T = 1$, should allow FPP to converge to a reasonable solution. We test this on CycleWorld (with Figure 7 in Appendix C), and find that for both larger mini-batch and number of updates FPP can get the error down to zero, whereas FPP without state updating cannot.

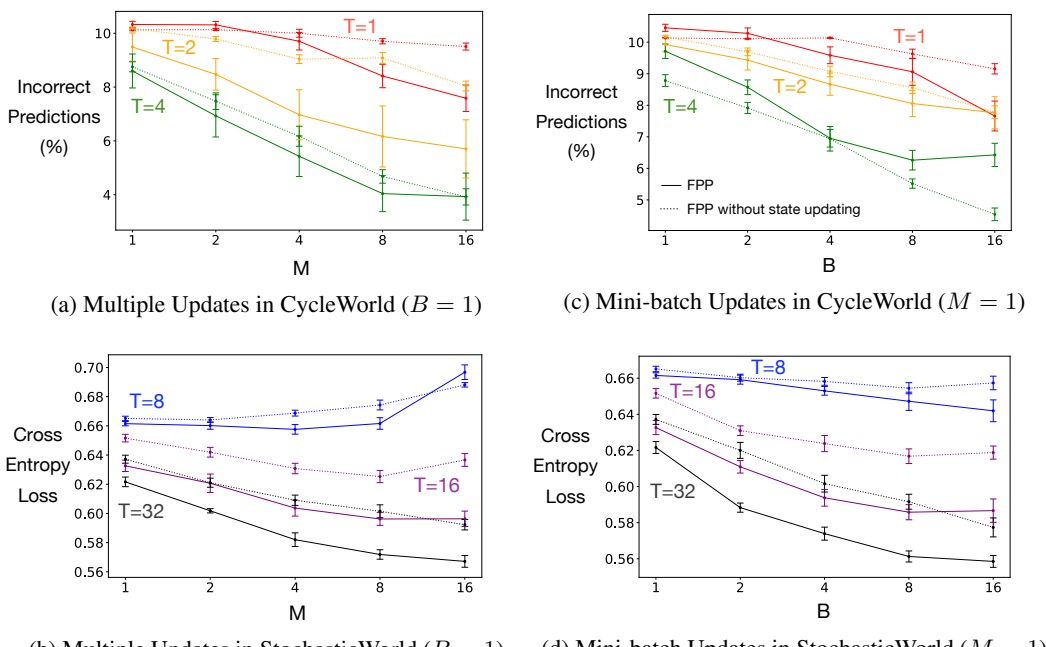

(a) Multiple Updates in CycleWorld ($B = 1$)  (c) Mini-batch Updates in CycleWorld ($M = 1$)

(b) Multiple Updates in StochasticWorld ($B = 1$)  (d) Mini-batch Updates in StochasticWorld ($M = 1$)

Figure 4: The performance for increase number of updates (with mini-batch of $B = 1$) and increasing mini-batch size (with number of updates $M = 1$). The numbers are average over 30 runs with 10000 training steps. The solid line is FPP and the dashed line is FPP without state updating.

## 6 CONCLUSION

The main objective of this paper is to reformulate RNN training to explicitly learn state variables. In particular, the goal is to investigate methods that can better distribute computation, and improve state updating without having to compute expensive—and potentially unstable—gradients back in time for each state. We introduce a new objective to explicitly learn state variables for RNNs, which breaks gradient dependence back in time. The choice of $T$ to compute gradients back in time is used only to improve training speed, rather than to effectively approximate gradients. We found that our algorithm, called FPP, was indeed more robust to $T$, than truncated BPTT was to its truncation level. We proved that our algorithm converges to a stationary point, under a fixed buffer, and so is a sound approach to using a buffer to train RNNs. Further, we chose simple optimization choices in this work; there are clear next steps for benefiting more from the decoupled update, such as by parallelizing updates across state variables. Overall, this work provides evidence that FPP could be a promising direction for robustly training RNNs, without the need to compute or approximate long gradients back in time.

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

# A  FULL PROOFS

## A.1  CONVERGENCE ON A FIXED BUFFER

At first glance, the update (9) is different than the update in Ghadimi *et al.* (2016, p. 276). Nevertheless, the following lemma guarantees that they are indeed the same.

**Lemma 1.** *Let $f$ be $L$ or a stochastic sample of $L$ and let $\alpha > 0$. Write $x = (\mathbf{W}, \boldsymbol{\beta}, \mathbf{S})$. Then*

$$\underset{u \in C}{\arg\min} \left\{ \langle \nabla f(x), u \rangle + \frac{1}{2\alpha} \|x - u\|^2 \right\} = \underset{u \in C}{\arg\min} \left\{ \|u - (x - \alpha \nabla f(x))\|^2 \right\} =: \Gamma(x - \alpha \nabla f(x)).$$

*Proof.* The proof is a straightforward calculation.

$$\underset{u \in C}{\arg\min} \left\{ \|u - (x - \alpha \nabla f(x))\|^2 \right\} = \underset{u \in C}{\arg\min} \left\{ \|u - x\|^2 + \alpha^2 \|\nabla f(x)\|^2 + 2\langle u - x, \alpha \nabla f(x) \rangle \right\}$$

$$= \underset{u \in C}{\arg\min} \left\{ \|u - x\|^2 + 2\alpha \langle u, \nabla f(x) \rangle \right\}$$

$$= \underset{u \in C}{\arg\min} \left\{ \langle \nabla f(x), u \rangle + \frac{1}{2\alpha} \|x - u\|^2 \right\}$$

$\square$

Our goal is to apply Corollary 3 of Ghadimi *et al.* (2016, p. 282). We must show that $\nabla g$ is Lipschitz on $C$ and demonstrate that Assumption **A1** in Ghadimi *et al.* (2016, p. 268) holds.

**Lemma 2.** $\nabla L$ *is Lipschitz on $C$.*

*Proof.* A function is Lipschitz if it gradient is bounded. Since $L$ is smooth and $C$ is compact (continuous functions on compact sets are bounded), this lemma follows. $\square$

**Lemma 3.** $\nabla L_k$ *is an unbiased estimate of $\nabla L$, where $k \sim$ uniform-$(T, n)$.*

*Proof.* The terms in $\nabla L_k$ corresponding to the gradients of $\mathbf{W}$ and $\boldsymbol{\beta}$ are exactly $\nabla_{\mathbf{W}} L$ and $\nabla_{\boldsymbol{\beta}} L$ in expectation, given that $k \sim$ uniform-$(T, n)$.

Let us consider the gradient elements corresponding to the parameters $s_{0:n}$. For shorthand, define $[a : b] := \{a, a + 1, \cdots, b - 1, b\}$. Define $P_0 := [0 : n - T], P_1 := [T : n]$. If $j \in P_0$, then $s_j$ predicts future states. If $s_j \in P_1$, then $s_j$ is predicted by other states in the regularizer terms of $L$. Note that $P_0$ and $P_1$ are not disjoint. First, we calculate.

$$\nabla_{\mathbf{s}_j} L(\mathbf{W}, \boldsymbol{\beta}, \mathbf{S}) := \begin{cases} \frac{1}{n-T+1} (\nabla_{\hat{\mathbf{s}}_{j+T}} \ell_{\boldsymbol{\beta}}(\hat{\mathbf{s}}_{j+T}; y_{j+T}) - \lambda(\mathbf{s}_{j+T} - \hat{\mathbf{s}}_{j+T}))^\top \nabla_{\mathbf{s}_j} \hat{\mathbf{s}}_j & \text{if } j \in P_0 \cap P_1^{\complement} \\ \frac{1}{n-T+1} \lambda(\mathbf{s}_j - \hat{\mathbf{s}}_j) & \text{if } j \in P_0^{\complement} \cap P_1 \\ \frac{1}{n-T+1} [\lambda(\mathbf{s}_j - \hat{\mathbf{s}}_j) + (\nabla_{\hat{\mathbf{s}}_{j+T}} \ell_{\boldsymbol{\beta}}(\hat{\mathbf{s}}_{j+T}; y_{j+T}) - \lambda(\mathbf{s}_{j+T} - \hat{\mathbf{s}}_{j+T}))^\top \nabla_{\mathbf{s}_j} \hat{\mathbf{s}}_j] \\ \qquad \text{if } j \in P_0 \cap P_1 \\ 0 & \text{if } j \in P_0^{\complement} \cap P_1^{\complement} \end{cases}$$

If $j \in P_0 \cap P_1^{\complement}$, then $\mathbf{s}_j$ does not show up as the target (i.e., the term that is not $\hat{\mathbf{s}}_k$) in any regularizer term of $L$. Hence, $\nabla_{\mathbf{s}_j} L_k$ is zero with probability $1 - \frac{1}{n-T+1}$, and is $(\nabla_{\hat{\mathbf{s}}_{j+T}} \ell_{\boldsymbol{\beta}}(\hat{\mathbf{s}}_{j+T}; y_{j+T}) - \lambda(\mathbf{s}_{j+T} - \hat{\mathbf{s}}_{j+T}))^\top \nabla_{\mathbf{s}_j} \hat{\mathbf{s}}_j$ with probability $\frac{1}{n-T+1}$.

If $j \in P_0^{\complement} \cap P_1$, then $s_j$ only shows up as a target in a regularizer term, so $\nabla_{\mathbf{s}_j} L_k$ is zero with probability $1 - \frac{1}{n-T+1}$ and is otherwise $\frac{1}{n-T+1} \lambda(\mathbf{s}_j - \hat{\mathbf{s}}_j)$.

If $j \in P_0 \cap P_1$, then $\nabla_{\mathbf{s}_j} L_k$ is zero with probability $1 - \frac{2}{n-T+1}$, $\lambda(\mathbf{s}_j - \hat{\mathbf{s}}_j)$ with probability $\frac{1}{n-T+1}$, and $(\nabla_{\hat{\mathbf{s}}_{j+T}} \ell_{\boldsymbol{\beta}}(\hat{\mathbf{s}}_{j+T}; y_{j+T}) - \lambda(\mathbf{s}_{j+T} - \hat{\mathbf{s}}_{j+T}))^\top \nabla_{\mathbf{s}_j} \hat{\mathbf{s}}_j$ with probability $\frac{1}{n-T+1}$.

The case for $j \in P_0^{\complement} \cap P_1^{\complement}$ is trivial. Consequently, $\mathbb{E}[\nabla_{\mathbf{s}_j} L_k] = \nabla_{\mathbf{s}_j} L$ for all $j \in \{0, \cdots, n\}$.

$\square$

**Lemma 4.** *The variance of $\nabla L_k$ is bounded on $C$.*

*Proof.* This follows because $\nabla L_k$ and $\nabla L$ are both smooth functions on a compact set $C$, and thus bounded. $\qquad\square$

**Theorem 1.** *Let $D$ be a Lipschitz constant of $\nabla L(\mathbf{W}, \boldsymbol{\beta}, \mathbf{S})$. Define probability mass functions*

$$P_N(k) := \frac{\alpha_k - D\alpha_k^2}{\sum_{j=1}^{N} \alpha_j - D\alpha_j^2}.$$

*for each $N \in \mathbb{N}$. Let $R$ be distributed according to $P_N$. Assume $\alpha_t = \frac{1}{2D}$ for all $t$ and that we perform $N$ stochastic updates. Write $x_R = (\mathbf{W}_R, \boldsymbol{\beta}_R, \mathbf{S}_R)$. Then*

$$\mathbb{E}\left[\frac{1}{\alpha_R^2}\|\Gamma(\alpha_R \nabla L(x_R))\|^2\right] = \mathcal{O}\left(\frac{1}{N}\right).$$

*Proof.* The $g_{X,R}$ (defined in Ghadimi *et al.* (2016, p. 271, 274)) in Corollary 3 of Ghadimi *et al.* (2016, p. 282) corresponds in our case to the following.

$$
\begin{aligned}
g_{X,R} &:= \frac{1}{\alpha_R}\left(x_R - \arg\min_{u \in C}\left\{\langle \nabla f(x), u\rangle + \frac{1}{2\alpha}\|x - u\|^2\right\}\right) \\
&= \frac{1}{\alpha_R}(x_R - \Gamma(x_R - \alpha_R \nabla L(x_R))).
\end{aligned}
$$

In the last line, we use Lemma 1. Since we project based on squared norm distance in (8) (corresponding to $\omega(x) = \frac{1}{2}\|x\|_2^2$ in Ghadimi *et al.* (2016)), the $\alpha$ in Ghadimi *et al.* (2016, p. 271) (not our step-size $\alpha_t$) can be set to 1.

After applying our Lemma 2, Lemma 3, and Lemma 4, we have from Corollary 3 of Ghadimi *et al.* (2016, p. 282) that

$$\mathbb{E}\left[\frac{1}{\alpha_R^2}\|\Gamma(x_R - \alpha_R \nabla L(x_R)) - x_R\|^2\right] = \mathcal{O}\left(\frac{1}{N}\right).$$

The only thing left to check is that $\Gamma(x_R - \alpha_R \nabla L(x_R)) - x_R = \Gamma(\nabla L(x_R))$.

$$
\begin{aligned}
\Gamma(x_R - \alpha_R \nabla L(x_R)) - x_R &= \arg\min_{u \in C}\left\{\|u - (x_R - \alpha_R \nabla L(x_R))\|^2\right\} - x_R \\
&= \arg\min_{u \in C}\left\{\|u - \nabla L(x_R))\|^2\right\} \\
&= \Gamma(\alpha_R \nabla L(x_R)).
\end{aligned}
$$

$\qquad\square$

## A.2 Recovery of RNN Solutions

Recall our goal is to compare to the RNN solutions of (10).

$$E(\mathbf{W}, \boldsymbol{\beta}, \mathbf{s}_0) := \sum_{i=1}^{n} \ell_{\boldsymbol{\beta}}(f_{\mathbf{W}}(\cdots f_{\mathbf{W}}(f_{\mathbf{W}}(\mathbf{s}_0, \mathbf{o}_1), \mathbf{o}_2), \cdots, \mathbf{o}_i); \mathbf{y}_i) \qquad \text{(10 revisited)}$$
$$\min_{\boldsymbol{\beta}, \mathbf{W}, \mathbf{s}_0} E(\mathbf{W}, \boldsymbol{\beta}, \mathbf{s}_0),$$

Let us also write a constrained version of the above problem, which we will use in the analysis of FPP.

$$E_{fpp}(\mathbf{W}, \boldsymbol{\beta}, \mathbf{s}_0, \cdots, \mathbf{s}_n) := \sum_{i=1}^{n} \ell_{\boldsymbol{\beta}}(f_{\mathbf{W}}(\mathbf{s}_{i-1}, \mathbf{o}_i); \mathbf{y}_i) \qquad (12)$$
$$\text{s.t. } \forall 1 \le i \le n, f_{\mathbf{W}}(\mathbf{s}_{i-1}, \mathbf{o}_i) = \mathbf{s}_i$$
$$\min_{\mathbf{W}, \boldsymbol{\beta}, \mathbf{s}_{0:n}} E_{fpp}(\mathbf{W}, \boldsymbol{\beta}, \mathbf{s}_0, \cdots, \mathbf{s}_n)$$

The idea is that FPP can be viewed as a way to solve the problem (12) and thus (10) through quadratic regularization.

We will use $\mathbf{s}_{0:n}$ as shorthand for $\{\mathbf{s}_0, \cdots, \mathbf{s}_n\}$, which in the main paper we labeled as $\mathbf{S}$, but for this proof it will be convenient to use explicit variables. Define the feasible set of (12) as

$$\Omega := \{(\mathbf{W}, \boldsymbol{\beta}, \mathbf{s}_{0:n}) : \mathbf{W} \in \mathbb{R}^w; \mathbf{s}_i \in \mathbb{R}^k; \boldsymbol{\beta} \in \mathbb{R}^b; \forall 1 \leq i \leq n, \, \mathbf{s}_i = f_{\mathbf{W}}(\mathbf{s}_{i-1}, \mathbf{o}_i)\}.$$

**Proposition 1.** *Let* $(\mathbf{W}^*, \boldsymbol{\beta}^*, \mathbf{s}_0^*)$ *be a local min of* (10). *For* $1 \leq i \leq n$, *define recursively* $\mathbf{s}_i^* := f_{\mathbf{W}^*}(\mathbf{s}_{i-1}^*, \mathbf{o}_i)$. *Then* $(\mathbf{W}^*, \boldsymbol{\beta}^*, \mathbf{s}_{0:n}^*)$ *is a local min of* (12).

*Let* $(\mathbf{W}^*, \boldsymbol{\beta}^*, \mathbf{s}_{0:n}^*)$ *be a local min of* (12). *Then* $(\mathbf{W}^*, \boldsymbol{\beta}^*, \mathbf{s}_0^*)$ *is a local min of* (10).

*Proof.* First, let $N \subset \mathbb{R}^{w+b+k}$ be a neighbourhood of $(\mathbf{W}^*, \boldsymbol{\beta}^*, \mathbf{s}_0^*)$ such that $\forall (\mathbf{W}, \boldsymbol{\beta}, \mathbf{s}_0) \in N$, we have

$$E(\mathbf{W}^*, \boldsymbol{\beta}^*, \mathbf{s}_0^*) \leq E(\mathbf{W}, \boldsymbol{\beta}, \mathbf{s}_0).$$

Without loss of generality, we may take $N$ to be open. Otherwise, by definition of a neighbourhood, we may take a smaller open set around $(\mathbf{W}^*, \boldsymbol{\beta}^*, \mathbf{s}_0^*)$ by definition of a neighbourhood and call that set $N$.

Let $\mathbf{s}_i^*$ be defined as above. Define $M := N \times \mathbb{R}^{nk}$, which is an open neighbourhood of $(\mathbf{W}^*, \boldsymbol{\beta}^*, \mathbf{s}_0^*)$ since $N$ is open. Let $(\mathbf{W}, \boldsymbol{\beta}, \mathbf{s}_{0:n}) \in M \cap \Omega$. Note that $(\mathbf{W}, \boldsymbol{\beta}, \mathbf{s}_0) \in N$. By definition of $\Omega$, we have that $f_{\mathbf{W}}(\mathbf{s}_{i-1}, \mathbf{o}_i) = \mathbf{s}_i$. Hence, $E_{fpp}(\mathbf{W}, \boldsymbol{\beta}, \mathbf{s}_{0:n}) = E(\mathbf{W}, \boldsymbol{\beta}, \mathbf{s}_0)$.

By definition of (12), (10), we have $E(\mathbf{W}^*, \boldsymbol{\beta}^*, \mathbf{s}_0^*) = E_{fpp}(\mathbf{W}^*, \boldsymbol{\beta}^*, \mathbf{s}_{0:n}^*)$. Finally,

$$E_{fpp}(\mathbf{W}^*, \boldsymbol{\beta}^*, \mathbf{s}_{0:n}^*) = E(\mathbf{W}^*, \boldsymbol{\beta}^*, \mathbf{s}_0^*) \leq E(\mathbf{W}, \boldsymbol{\beta}, \mathbf{s}_0) = E_{fpp}(\mathbf{W}, \boldsymbol{\beta}, \mathbf{s}_{0:n}).$$

For the second part of the proof, assume $(\mathbf{W}^*, \boldsymbol{\beta}^*, \mathbf{s}_{0:n}^*)$ is a local min of (12), meaning there is a neigbourhood $M \subset \mathbb{R}^{w+b+(n+1)k}$ of $(\mathbf{W}^*, \boldsymbol{\beta}^*, \mathbf{s}_{0:n}^*)$ such that for every $(\mathbf{W}, \boldsymbol{\beta}, \mathbf{s}_{0:n}) \in M \cap \Omega$,

$$E_{fpp}(\mathbf{W}^*, \boldsymbol{\beta}^*, \mathbf{s}_{0:n}^*) \leq E_{fpp}(\mathbf{W}, \boldsymbol{\beta}, \mathbf{s}_{0:n}).$$

Similarly, without loss of generality, we can assume that $M$ is an open ball, so we may write for some $\epsilon > 0$, $M = B_\epsilon(\mathbf{W}^*, \boldsymbol{\beta}^*, \mathbf{s}_{0:n}^*)$.

We will construct an open set $N \subset \mathbb{R}^{w+b+k}$ such that $(\mathbf{W}^*, \boldsymbol{\beta}^*, \mathbf{s}_0^*)$ is a local min with respect to $N$. Define the projection $\pi$ onto the first $w + b + k$ indices. Define $N := \pi(M \cap \Omega)$. Let us show that $N$ is open.

We will write $f_{\mathbf{W}}(\mathbf{s}_{0:n-1}, \mathbf{o}_{1:n})$ to mean $\{f_{\mathbf{W}}(\mathbf{s}_0, \mathbf{o}_1), \cdots, f_{\mathbf{W}}(f_{\mathbf{W}}(\cdots (\mathbf{s}_0, \mathbf{o}_1), \mathbf{o}_2), \cdots, \mathbf{o}_n))\}$. We can write $N$ as

$$\begin{aligned} N &= \{(\mathbf{W}, \boldsymbol{\beta}, \mathbf{s}_0) : (\mathbf{W}, \boldsymbol{\beta}, \mathbf{s}_{0:n}) \in \Omega \cap B_\epsilon(\mathbf{W}^*, \boldsymbol{\beta}^*, \mathbf{s}_{0:n}^*)\} \\ &= \{(\mathbf{W}, \boldsymbol{\beta}, \mathbf{s}_0) : (\mathbf{W}, \boldsymbol{\beta}, \mathbf{s}_0, f_{\mathbf{W}}(\mathbf{s}_{0:n-1}, \mathbf{o}_{1:n})) \in B_\epsilon(\mathbf{W}^*, \boldsymbol{\beta}^*, \mathbf{s}_{0:n}^*)\} \\ &= \{(\mathbf{W}, \boldsymbol{\beta}, \mathbf{s}_0) : \|(\mathbf{W}, \boldsymbol{\beta}, \mathbf{s}_0, f_{\mathbf{W}}(\mathbf{s}_{0:n-1}, \mathbf{o}_{1:n})) - (\mathbf{W}^*, \boldsymbol{\beta}^*, s_{0:n}^*)\| < \epsilon\} \end{aligned}$$

On the second line, we used the fact that $\mathbf{s}_i = f_{\mathbf{W}}(\mathbf{s}_{i-1}, \mathbf{o}_i)$ in $\Omega$. Since the norm and $f$ are continuous and $(-\infty, \epsilon)$ is open, we have that $N$, a continuous preimage of an open set, is open.

Now, let $(\mathbf{W}, \boldsymbol{\beta}, \mathbf{s}_0) \in N$ such that $\exists \mathbf{s}_{1:n}$ with $(\mathbf{W}, \boldsymbol{\beta}, \mathbf{s}_{0:n}) \in M \cap \Omega$.

$$E(\mathbf{W}^*, \boldsymbol{\beta}^*, \mathbf{s}_0^*) = E_{fpp}(\mathbf{W}^*, \boldsymbol{\beta}^*, \mathbf{s}_{0:n}^*) \leq E_{fpp}(\mathbf{W}, \boldsymbol{\beta}, \mathbf{s}_{0:n}) = E(\mathbf{W}, \boldsymbol{\beta}, \mathbf{s}_0)$$

The claim follows. $\square$

**Proposition 2.** *The first order KKT equations for* (10) *and for* (12) *are the same.*

*Proof.* Given $(\mathbf{W}^*, \boldsymbol{\beta}^*, \mathbf{s}_0^*)$, for $1 \leq i \leq n$ define $\tilde{s}_i := f_{\mathbf{W}}(\tilde{s}_{i-1}, \mathbf{o}_i)$, where $\tilde{s}_0 := \mathbf{s}_0^*$. If we write $\frac{\partial f_{\mathbf{W}^*}(\tilde{s}_l, \mathbf{o}_{l+1})}{\partial \mathbf{s}_l}$ for instance, this is taken to mean the gradient of $f_{\mathbf{W}^*}(\tilde{s}_l, \mathbf{o}_{l+1})$ with respect to the function arguments corresponding to $\tilde{s}_l$. Furthermore, when writing $\frac{\partial f_{\mathbf{W}^*}(\tilde{s}_j, \mathbf{o}_{j+1})}{\partial \mathbf{W}}$, we only mean the gradient with respect to the parameters of the outer $f_{\mathbf{W}^*}$, and not with respect to any of the parameters of $\tilde{s}_j$.

Using the chain rule for the first and third equations below, the first order KKT conditions for (10) are given by

$$\frac{\partial E(\mathbf{W}^*, \boldsymbol{\beta}^*, \mathbf{s}_0^*)}{\partial \mathbf{W}} = \frac{\partial \ell_{\boldsymbol{\beta}^*}(f_{\mathbf{W}^*}(\mathbf{s}_0^*, \mathbf{o}_1); y_1)}{\partial f_{\mathbf{W}}} \frac{\partial f_{\mathbf{W}^*}(\mathbf{s}_0^*, \mathbf{o}_1)}{\partial \mathbf{W}} + \tag{13}$$

$$\sum_{j=1}^{n-1} \frac{\partial \ell_{\boldsymbol{\beta}^*}(f_{\mathbf{W}^*}(\tilde{s}_j, \mathbf{o}_{j+1}); y_{j+1})}{\partial f_{\mathbf{W}}}$$

$$\left( \frac{\partial f_{\mathbf{W}^*}(\tilde{s}_j, \mathbf{o}_{j+1})}{\partial \mathbf{W}} + \sum_{i=1}^{j} \prod_{l=i}^{j} \frac{\partial f_{\mathbf{W}^*}(\tilde{s}_l, \mathbf{o}_{l+1})}{\partial \mathbf{s}_l} \right) \frac{\partial f_{\mathbf{W}^*}(\tilde{s}_{i-1}, \mathbf{o}_i)}{\partial \mathbf{W}}$$

$$= 0$$

$$\frac{\partial E(\mathbf{W}^*, \boldsymbol{\beta}^*, \mathbf{s}_0^*)}{\partial \boldsymbol{\beta}} = \sum_{i=1}^{n} \frac{\partial \ell_{\boldsymbol{\beta}^*}(f_{\mathbf{W}^*}(\tilde{s}_{i-1}, \mathbf{o}_i); \mathbf{y}_i)}{\partial \boldsymbol{\beta}} = 0 \tag{14}$$

$$\frac{\partial E(\mathbf{W}^*, \boldsymbol{\beta}^*, \mathbf{s}_0^*)}{\partial \mathbf{s}_0} = \left( \frac{\partial \ell_{\boldsymbol{\beta}^*}(f_{\mathbf{W}^*}(\mathbf{s}_0^*, \mathbf{o}_1); y_1)}{\partial f_{\mathbf{W}}} + \left( \sum_{i=1}^{n-1} \frac{\partial \ell_{\boldsymbol{\beta}^*}(f_{\mathbf{W}^*}(\tilde{s}_i, \mathbf{o}_{i+1}); y_{i+1})}{\partial f_{\mathbf{W}}} \right. \right. \tag{15}$$

$$\left. \left. \prod_{l=1}^{i} \frac{\partial f_{\mathbf{W}^*}(\tilde{s}_l, \mathbf{o}_{l+1})}{\partial \mathbf{s}_l} \right) \right) \frac{\partial f_{\mathbf{W}^*}(\mathbf{s}_0^*, \mathbf{o}_1)}{\partial \mathbf{s}_0}$$

$$= 0$$

The Lagrangian for (12) is

$$\mathscr{L}_{fpp}(\mathbf{W}, \boldsymbol{\beta}, \mathbf{s}_{0:n}) = \sum_{i=1}^{n} \ell_{\boldsymbol{\beta}}(f_{\mathbf{W}}(\mathbf{s}_{i-1}, \mathbf{o}_i); \mathbf{y}_i) - \lambda_i^T(f_{\mathbf{W}}(\mathbf{s}_{i-1}, \mathbf{o}_i) - \mathbf{s}_i), \tag{16}$$

where $\lambda_i \in \mathbb{R}^k$ for $1 \le i \le n$ are Lagrange multipliers. We define $\lambda_0 := 0$ for convenience. The KKT equations for (16) are

$$\frac{\partial \mathscr{L}_{fpp}(\mathbf{W}^*, \boldsymbol{\beta}^*, \mathbf{s}_{0:n}^*)}{\partial \mathbf{W}} = \sum_{i=1}^{n} \frac{\partial \ell_{\boldsymbol{\beta}^*}(f_{\mathbf{W}^*}(\mathbf{s}_{i-1}^*, \mathbf{o}_i); \mathbf{y}_i)}{\partial f_{\mathbf{W}}} \frac{\partial f_{\mathbf{W}^*}(\mathbf{s}_{i-1}^*, \mathbf{o}_i)}{\partial \mathbf{W}} \tag{17}$$

$$- \lambda_i^T \frac{\partial f_{\mathbf{W}^*}(\mathbf{s}_{i-1}^*, \mathbf{o}_i)}{\partial \mathbf{W}} = 0$$

$$\frac{\partial \mathscr{L}_{fpp}(\mathbf{W}^*, \boldsymbol{\beta}^*, \mathbf{s}_{0:n}^*)}{\partial \boldsymbol{\beta}} = \sum_{i=1}^{n} \frac{\partial \ell_{\boldsymbol{\beta}^*}(f_{\mathbf{W}^*}(\mathbf{s}_{i-1}^*, \mathbf{o}_i); \mathbf{y}_i)}{\partial \boldsymbol{\beta}} = 0$$

$$\frac{\partial \mathscr{L}_{fpp}(\mathbf{W}^*, \boldsymbol{\beta}^*, \mathbf{s}_{0:n}^*)}{\partial \mathbf{s}_j} = \begin{cases} \lambda_n^T & \text{if } j = n \\ \lambda_j^T + \left( \frac{\partial \ell_{\boldsymbol{\beta}^*}(f_{\mathbf{W}^*}(\mathbf{s}_j^*, \mathbf{o}_{j+1}); y_{j+1})}{\partial f_{\mathbf{W}}} - \lambda_{j+1}^T \right) \frac{\partial f_{\mathbf{W}^*}(\mathbf{s}_j^*, \mathbf{o}_{j+1})}{\partial \mathbf{s}_j} \\ \qquad \text{if } 0 \le j < n \end{cases} \tag{18}$$

$$= 0$$

$$\mathbf{s}_i^* = f_{\mathbf{W}^*}(\mathbf{s}_{i-1}^*, \mathbf{o}_i), \quad \forall \, 1 \le i \le n$$

First, let us find a closed-form expression for $\lambda_i$.

**Lemma 1.** *Let $0 \le j \le n$. Then*

$$\lambda_j^T = - \left( \sum_{i=j}^{n-1} \frac{\partial \ell_{\boldsymbol{\beta}^*}(f_{\mathbf{W}^*}(\mathbf{s}_i^*, \mathbf{o}_{i+1}); y_{i+1})}{\partial f_{\mathbf{W}}} \prod_{l=j}^{i} \frac{\partial f_{\mathbf{W}^*}(\mathbf{s}_l^*, \mathbf{o}_{l+1})}{\partial \mathbf{s}_l} \right)$$

*Proof.* We proceed by induction. The base case and the case $j = n - 1$ are trivial. Assume the claim is true for $m + 1 > 0$. We will show the claim for $j = m$. Using the KKT equations (17) and the induction hypothesis,

$$\lambda_m^T := -\left(\frac{\partial \ell_{\boldsymbol{\beta}^*}(f_{\mathbf{W}^*}(\mathbf{s}_m^*, \mathbf{o}_{m+1}); y_{m+1})}{\partial f_{\mathbf{W}}} - \lambda_{m+1}^T\right)\frac{\partial f_{\mathbf{W}^*}(\mathbf{s}_m^*, \mathbf{o}_{m+1})}{\partial \mathbf{s}_m}$$

$$= -\left(\frac{\partial \ell_{\boldsymbol{\beta}^*}(f_{\mathbf{W}^*}(\mathbf{s}_m^*, \mathbf{o}_{m+1}); y_{m+1})}{\partial f_{\mathbf{W}}} + \sum_{i=m+1}^{n-1}\frac{\partial \ell_{\boldsymbol{\beta}^*}(f_{\mathbf{W}^*}(\mathbf{s}_i^*, \mathbf{o}_{i+1}); y_{i+1})}{\partial f_{\mathbf{W}}}\right.$$

$$\left.\prod_{l=m+1}^{i}\frac{\partial f_{\mathbf{W}^*}(\mathbf{s}_l^*, \mathbf{o}_{l+1})}{\partial \mathbf{s}_l}\right)\frac{\partial f_{\mathbf{W}^*}(\mathbf{s}_m^*, \mathbf{o}_{m+1})}{\partial \mathbf{s}_m}$$

$$= -\frac{\partial \ell_{\boldsymbol{\beta}^*}(f_{\mathbf{W}^*}(\mathbf{s}_m^*, \mathbf{o}_{m+1}); y_{m+1})}{\partial f_{\mathbf{W}}}\frac{\partial f_{\mathbf{W}^*}(\mathbf{s}_m^*, \mathbf{o}_{m+1})}{\partial \mathbf{s}_m} -$$

$$\sum_{i=m+1}^{n-1}\frac{\partial \ell_{\boldsymbol{\beta}^*}(f_{\mathbf{W}^*}(\mathbf{s}_i^*, \mathbf{o}_{i+1}); y_{i+1})}{\partial f_{\mathbf{W}}}\prod_{l=m+1}^{i}\frac{\partial f_{\mathbf{W}^*}(\mathbf{s}_l^*, \mathbf{o}_{l+1})}{\partial \mathbf{s}_l}\frac{\partial f_{\mathbf{W}^*}(\mathbf{s}_m^*, \mathbf{o}_{m+1})}{\partial \mathbf{s}_m}$$

$$= -\left(\sum_{i=m}^{n-1}\frac{\partial \ell_{\boldsymbol{\beta}^*}(f_{\mathbf{W}^*}(\mathbf{s}_i^*, \mathbf{o}_{i+1}); y_{i+1})}{\partial f_{\mathbf{W}}}\prod_{l=m}^{i}\frac{\partial f_{\mathbf{W}^*}(\mathbf{s}_l^*, \mathbf{o}_{l+1})}{\partial \mathbf{s}_l}\right)$$

$\square$

Now, we will show that the sets of equations are the same. First, it is clear that the two equations involving gradients of $\boldsymbol{\beta}$ in (13) and (17) are the same given that the constraint must be satisfied in (17). Now consider the equations involving gradients with respect to $\mathbf{W}$.

$$\frac{\partial \mathscr{L}_{fpp}}{\partial \mathbf{W}} = \sum_{i=1}^{n}\frac{\partial \ell_{\boldsymbol{\beta}^*}(f_{\mathbf{W}^*}(\mathbf{s}_{i-1}^*, \mathbf{o}_i); \mathbf{y}_i)}{\partial f_{\mathbf{W}}}\frac{\partial f_{\mathbf{W}^*}(\mathbf{s}_{i-1}^*, \mathbf{o}_i)}{\partial \mathbf{W}} +$$

$$\sum_{i=1}^{n}\left(\sum_{j=i}^{n-1}\frac{\partial \ell_{\boldsymbol{\beta}^*}(f_{\mathbf{W}^*}(\mathbf{s}_j^*, \mathbf{o}_{j+1}); y_{j+1})}{\partial f_{\mathbf{W}}}\prod_{l=i}^{j}\frac{\partial f_{\mathbf{W}^*}(\mathbf{s}_l^*, \mathbf{o}_{l+1})}{\partial \mathbf{s}_l}\right)\frac{\partial f_{\mathbf{W}^*}(\mathbf{s}_{i-1}^*, \mathbf{o}_i)}{\partial \mathbf{W}}$$

$$= \sum_{j=1}^{n}\frac{\partial \ell_{\boldsymbol{\beta}^*}(f_{\mathbf{W}^*}(\mathbf{s}_{j-1}^*, \mathbf{o}_j); y_j)}{\partial f_{\mathbf{W}}}\frac{\partial f_{\mathbf{W}^*}(\mathbf{s}_{j-1}^*, \mathbf{o}_j)}{\partial \mathbf{W}} +$$

$$\sum_{j=1}^{n-1}\left(\sum_{i=1}^{j}\frac{\partial \ell_{\boldsymbol{\beta}^*}(f_{\mathbf{W}^*}(\mathbf{s}_j^*, \mathbf{o}_{j+1}); y_{j+1})}{\partial f_{\mathbf{W}}}\prod_{l=i}^{j}\frac{\partial f_{\mathbf{W}^*}(\mathbf{s}_l^*, \mathbf{o}_{l+1})}{\partial \mathbf{s}_l}\right)\frac{\partial f_{\mathbf{W}^*}(\mathbf{s}_{i-1}^*, \mathbf{o}_i)}{\partial \mathbf{W}}$$

$$= \frac{\partial \ell_{\boldsymbol{\beta}^*}(f_{\mathbf{W}^*}(\mathbf{s}_0^*, \mathbf{o}_1); y_1)}{\partial f_{\mathbf{W}}}\frac{\partial f_{\mathbf{W}^*}(\mathbf{s}_0^*, \mathbf{o}_1)}{\partial \mathbf{W}} + \sum_{j=1}^{n-1}\frac{\partial \ell_{\boldsymbol{\beta}^*}(f_{\mathbf{W}^*}(\mathbf{s}_j^*, \mathbf{o}_{j+1}); y_{j+1})}{\partial f_{\mathbf{W}}}$$

$$\left(\frac{\partial f_{\mathbf{W}^*}(\mathbf{s}_j^*, \mathbf{o}_{j+1})}{\partial \mathbf{W}} + \sum_{i=1}^{j}\prod_{l=i}^{j}\frac{\partial f_{\mathbf{W}^*}(\mathbf{s}_l^*, \mathbf{o}_{l+1})}{\partial \mathbf{s}_l}\right)\frac{\partial f_{\mathbf{W}^*}(\mathbf{s}_{i-1}^*, \mathbf{o}_i)}{\partial \mathbf{W}}$$

$$= 0.$$

By substituting in the constraint equations $\mathbf{s}_i^* = f_{\mathbf{W}^*}(\mathbf{s}_{i-1}^*, \mathbf{o}_i)$, this recovers exactly the gradient with respect to $\mathbf{W}$ in (13).

Finally, consider the gradient with respect to $\mathbf{s}_0$.

$$
\begin{aligned}
\frac{\partial \mathscr{L}_{fpp}(\mathbf{W}^*, \boldsymbol{\beta}^*, \mathbf{s}_{0:n}^*)}{\partial \mathbf{s}_0} &= \lambda_0^T + \left( \frac{\partial \ell_{\boldsymbol{\beta}^*}(f_{\mathbf{W}^*}(\mathbf{s}_0^*, \mathbf{o}_1); y_1)}{\partial f_{\mathbf{W}}} - \lambda_1^T \right) \frac{\partial f_{\mathbf{W}^*}(\mathbf{s}_0^*, \mathbf{o}_1)}{\partial \mathbf{s}_0} \\
&= \left( \frac{\partial \ell_{\boldsymbol{\beta}^*}(f_{\mathbf{W}^*}(\mathbf{s}_0^*, \mathbf{o}_1); y_1)}{\partial f_{\mathbf{W}}} + \left( \sum_{i=1}^{n-1} \frac{\partial \ell_{\boldsymbol{\beta}^*}(f_{\mathbf{W}^*}(\mathbf{s}_i^*, \mathbf{o}_{i+1}); y_{i+1})}{\partial f_{\mathbf{W}}} \right. \right. \\
&\qquad \left. \left. \prod_{l=1}^{i} \frac{\partial f_{\mathbf{W}^*}(\mathbf{s}_l^*, \mathbf{o}_{l+1})}{\partial \mathbf{s}_l} \right) \right) \frac{\partial f_{\mathbf{W}^*}(\mathbf{s}_0^*, \mathbf{o}_1)}{\partial \mathbf{s}_0} \\
&= 0.
\end{aligned}
$$

This matches the corresponding equation in (13). $\qquad\square$

**Proposition 3.** *Let* $(\mathbf{W}^*, \boldsymbol{\beta}^*, \mathbf{s}_{0:n}^*)$ *be a local min of* (12). *Write the constraints of* (12) *as a vector:*

$$
h(\mathbf{W}, \mathbf{s}_{0:n}) := \left[ [(f_{\mathbf{W}}(\mathbf{s}_0, \mathbf{o}_1) - \mathbf{s}_1)^T \quad \cdots \quad (f_{\mathbf{W}}(\mathbf{s}_{n-1}, \mathbf{o}_n) - \mathbf{s}_n)^T \right] \tag{19}
$$

*Index each element of* $h$ *by* $h_i$. *Then the vectors* $\nabla h_i(\mathbf{W}^*, \mathbf{s}_{0:n}^*)$ *are linearly independent.*

*Proof.* In the following, we will write $\frac{\partial [g]_l}{\partial x_j^i}$ to mean the derivative of the $l$-th component of $g$ with respect to the $i$-th component of $x_j$. For compactness, write $g(i) := f_{\mathbf{W}^*}(\mathbf{s}_{i-1}^*, \mathbf{o}_i) - \mathbf{s}_i^*$. We can write the Jacobian $\nabla h(\mathbf{W}^*, \mathbf{s}_{0:n}^*)$ as

$$
\nabla h(\mathbf{W}^*, \mathbf{s}_{0:n}^*) = \begin{bmatrix}
\frac{\partial [g(1)]_1}{\partial \mathbf{W}^1} & \cdots & \frac{\partial [g(1)]_1}{\partial \mathbf{W}^w} & \frac{\partial [g(1)]_1}{\partial s_1^1} & \cdots & \frac{\partial [g(1)]_1}{\partial s_1^k} & \cdots & \frac{\partial [g(1)]_1}{\partial \mathbf{s}_n^k} \\
\vdots & \vdots & \vdots & \vdots & \vdots & \vdots & & \vdots \\
\frac{\partial [g(1)]_k}{\partial \mathbf{W}^1} & \cdots & \frac{\partial [g(1)]_k}{\partial \mathbf{W}^w} & \frac{\partial [g(1)]_k}{\partial s_1^1} & \cdots & \frac{\partial [g(1)]_k}{\partial s_1^k} & \cdots & \frac{\partial [g(1)]_k}{\partial \mathbf{s}_n^k} \\
\frac{\partial [g(2)]_1}{\partial \mathbf{W}^1} & \cdots & \frac{\partial [g(2)]_1}{\partial \mathbf{W}^w} & \frac{\partial [g(2)]_1}{\partial s_1^1} & \cdots & \frac{\partial [g(2)]_1}{\partial s_1^k} & \cdots & \frac{\partial [g(2)]_1}{\partial \mathbf{s}_n^k} \\
\vdots & \vdots & \vdots & \vdots & \vdots & \vdots & & \vdots \\
\frac{\partial [g(n)]_k}{\partial \mathbf{W}^1} & \cdots & \frac{\partial [g(n)]_k}{\partial \mathbf{W}^w} & \frac{\partial [g(n)]_k}{\partial s_1^1} & \cdots & \frac{\partial [g(n)]_k}{\partial s_1^k} & \cdots & \frac{\partial [g(n)]_k}{\partial \mathbf{s}_n^k}
\end{bmatrix}
$$

We will show that the rows of $\nabla h(\mathbf{W}^*, \mathbf{s}_{0:n}^*)$ are linearly independent. To this end, let $\lambda_{ij} \in \mathbb{R}$ for $i \in \{1, \cdots, n\}, j \in \{1, \cdots, k\}$ be such that:

$$
\sum_{j=1}^{k} \sum_{i=1}^{n} \lambda_{ij} \nabla [f_{\mathbf{W}^*}(\mathbf{s}_{i-1}^*, \mathbf{o}_i) - \mathbf{s}_i^*]_j = 0.
$$

In particular, for $1 \le a \le n, 1 \le b \le k$,

$$
\sum_{j=1}^{k} \sum_{i=1}^{n} \lambda_{ij} \frac{\partial [f_{\mathbf{W}^*}(\mathbf{s}_{i-1}^*, \mathbf{o}_i) - \mathbf{s}_i^*]_j}{\partial \mathbf{s}_a^b} = \sum_{j=1}^{k} \sum_{i=1}^{n} \lambda_{ij} \left( \delta_a^{i-1} \frac{\partial [f_{\mathbf{W}^*}(\mathbf{s}_{i-1}^*, \mathbf{o}_i)]_j}{\partial \mathbf{s}_a^b} - \delta_a^i \delta_b^j \right) \tag{20}
$$

$$
= 1_{a<n} \sum_{j=1}^{k} \lambda_{a+1,j} \frac{\partial [f_{\mathbf{W}^*}(\mathbf{s}_a^*, \mathbf{o}_{a+1})]_j}{\partial \mathbf{s}_{a+1}^b} - \lambda_{ab} \tag{21}
$$

$$
= 0 \tag{22}
$$

By setting $a = n$, we have that $\lambda_{nb} = 0$ for all $1 \le b \le k$. Setting $a = n-1$, we similarly have that $\lambda_{n-1,b} = 0$. Proceeding in this fashion, we have that $\lambda_{ab} = 0$ for all $1 \le a \le n, 1 \le b \le k$. Actually, we did not at any point use the fact that $(\mathbf{W}^*, \boldsymbol{\beta}^*, \mathbf{s}_{0:n}^*)$ is a local min, so that the constraint gradients are linearly independent everywhere, and in particular at $(\mathbf{W}^*, \boldsymbol{\beta}^*, \mathbf{s}_{0:n}^*)$. $\qquad\square$

**Theorem 2.** *Assume we have a positive, increasing sequence* $\{\lambda_k\} \to \infty$, *a non-negative sequence* $\{\epsilon_k\} \to 0$, *and a sequence of points* $\{(\mathbf{W}_k, \boldsymbol{\beta}_k, \mathbf{S}_k)\}$ *such that* $\|\nabla L(\mathbf{W}_k, \boldsymbol{\beta}_k, \mathbf{S}_k); \lambda_k)\| \leq \epsilon_k$ *for*

$$L(\mathbf{W}_k, \boldsymbol{\beta}_k, \mathbf{S}_k); \lambda_k) \stackrel{\text{def}}{=} \frac{1}{n} \sum_{i=1}^{n} \ell_{\boldsymbol{\beta}}(f_{\mathbf{W}}(\mathbf{s}_{i-1}, \mathbf{o}_i); \mathbf{y}_i) + \frac{\lambda_k}{2} \|\mathbf{s}_i - f_{\mathbf{W}}(\mathbf{s}_{i-1}, \mathbf{o}_i)\|_2^2 \qquad (11)$$

*Assume further that* $\{(\mathbf{W}_k, \boldsymbol{\beta}_k, \mathbf{S}_k)\}$ *has a convergent subsequence* $\{(\mathbf{W}_{k_i}, \boldsymbol{\beta}_{k_i}, \mathbf{S}_{k_i})\}$ *with limit* $(\mathbf{W}^*, \boldsymbol{\beta}^*, \mathbf{S}^*)$. *Then* $(\mathbf{W}^*, \boldsymbol{\beta}^*, \mathbf{S}^*)$ *is a KKT point of the constrained FPP objective (see* (12)*) and* $(\mathbf{W}^*, \boldsymbol{\beta}^*, \mathbf{s}_0^*)$ *is a KKT point of the RNN objective* (10)*. Further, if* $(\mathbf{W}^*, \boldsymbol{\beta}^*, \mathbf{S}^*)$ *is a local min of the constrained FPP objective, then* $(\mathbf{W}^*, \boldsymbol{\beta}^*, \mathbf{s}_0^*)$ *is a local min of* (10)*.*

*Proof.* By Proposition 3 and Proposition 2.3 from Bertsekas (1982), We have the existence of a Lagrange multiplier vector $\lambda$ such that

$$\nabla E_{fpp}(\mathbf{W}^*, \boldsymbol{\beta}^*, \mathbf{s}_{0:n}^*) - \nabla h(\mathbf{W}^*, \mathbf{s}_{0:n}^*)\lambda = 0,$$
$$h(\mathbf{W}^*, \mathbf{s}_{0:n}^*) = 0,$$

where $h(\mathbf{W}^*, \mathbf{s}_{0:n}^*)$ is as in Proposition 3. Hence, $(\mathbf{W}^*, \boldsymbol{\beta}^*, \mathbf{s}_{0:n}^*)$ is a KKT point of (12).

By Proposition 2, $(\mathbf{W}^*, \boldsymbol{\beta}^*, \mathbf{s}_0^*)$ is a KKT point for (10). Finally, if $(\mathbf{W}^*, \boldsymbol{\beta}^*, \mathbf{s}_{0:n}^*)$ is a local min of (12), then by Proposition 1 we have that $(\mathbf{W}^*, \boldsymbol{\beta}^*, \mathbf{s}_0^*)$ is a local min of (10). $\square$

## B   PARAMETER STUDY

We investigate the sensitivity of FPP to its two key parameters: the length of the trajectory $T$, and the buffer size $N$. Overall, the losses on y-axis of Figure 5 show that FPP is robust to buffer size and truncation length. As expected, for very small T, performance degrades, but otherwise the move from T= 10 to T= 50 does not result in a large difference. The algorithm was quite invariant to buffer size, starting from a reasonable size of 100. For too large a buffer with a small number of updates, performance did degrade somewhat. Overall, though, across this wide range of settings, FPP performed consistently well.

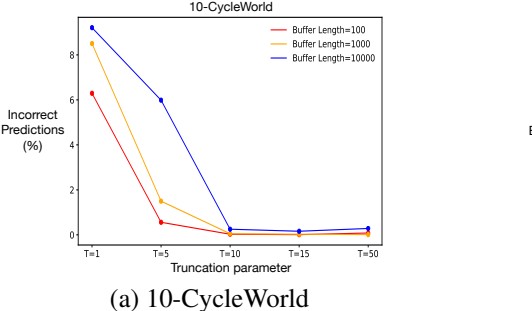
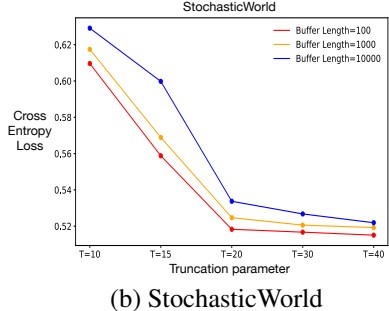

(a) 10-CycleWorld          (b) StochasticWorld

Figure 5: Sensitivity to buffer length and trajectory length in FPP, for buffer sizes 100, 1000 and 10000 and truncations of 1, 5,10,15 and 50.

We also investigated how performance changes when changing $\lambda$. Throughout all previous experiments, we simply set $\lambda = 1$, to avoid unfairly tuning our method to each problem. Interestingly, tuning $\lambda$ does enable further performance improvements, though the algorithm worked well for quite a large range of $\lambda$.

## C   EXPERIMENTAL DETAILS

The dynamics for the Stochastic World environment are in Table 1.

For all experiments, we use RMSprop optimizer and the learning rate is chosen over the set $\{0.0001, 0.0003, 0.001, 0.003, 0.01, 0.03\}$ based on the average accuracy/loss. For real datasets, we use multiple trajectories to speed up training. The details of each task are provided below:

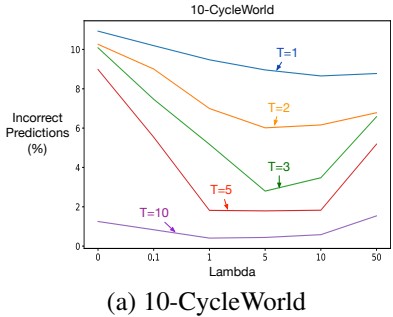
(a) 10-CycleWorld

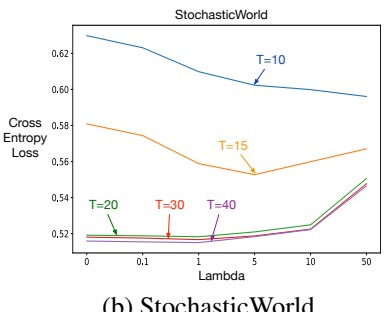
(b) StochasticWorld

Figure 6: Sensitivity of Lambda for various values of T. For small T, higher lambda works better suggesting the impact of propagation of state values across the buffer.

| $P(Y_t = 1 \vert O_{t-T_1}, O_{t-T_2})$ | $O_{t-T_1}$ | $O_{t-T_2}$ |
|:---:|:---:|:---:|
| 50% | 0 | 0 |
| 100% | 1 | 0 |
| 25% | 0 | 1 |
| 75% | 1 | 1 |

Table 1: The conditional probability of the target output given the past observations.

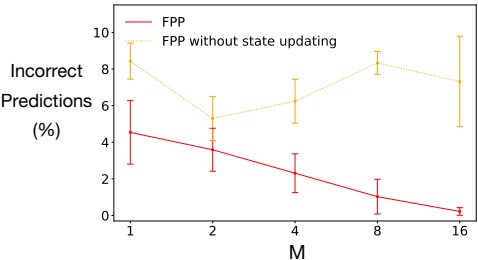

Figure 7: The performance of FPP and FPP without state updating with $T = 1$, $B = 16$ after 50000 training steps, for varying $M$. This result highlights that FPP can better take advantage of more updates and larger mini-batches, with its sound updating strategy on a buffer.

## C.1 CYCLEWORLD

Network Type = simple RNN
Hidden Units = 4

## C.2 STOCHASTIC WORLD

Network Type = simple RNN
Hidden Units = 32

## C.3 SEQUENTIAL MNIST

Network Type = simple RNN
Hidden Units = 512
Image Size = 784 pixels
Input Dimension = 28 pixels
Number of Steps = 28000 (1000 images of 28 steps)

Number of Trajectories = 20

## C.4   PTB

Network Type = LSTM
Hidden Units = 200
Vocabulary Size = 10000
Embedding Size = 200
Number of Steps = 5000 (5000 samples in the dataset)
Number of Trajectories = 20

