# OpenReview forum: "Training Recurrent Neural Networks Online by Learning Explicit State Variables"
_ICLR.cc/2020/Conference — Accept (Poster)_

### Official Review · AnonReviewer1 · 2019-10-20
**Official Blind Review #1**

**Rating:** 6

**Review:**


Background: The authors consider the problem of training RNNs in an online fashion. The authors note that RNNs are trained using BPTT, which prevents them from being trained in an online fashion.  There have been various approximations which has been proposed which are based on RTRL or approximations to RTRL, as current approximations based on RTRL has high computational complexity.

Proposed Method: The authors propose to learn the state of the RNNs explicitly by improving the prediction accuracy at each time step as well as predicting the "next" state of the RNN. The authos note that the constraint of predicting the next state  is a fixed-point formula for the states underthe given RNN dynamics.

Clarity of the paper: The paper is clearly written.

Related work : Most of the relevant related work has been covered in the paper and discussed. I like it. These two related work could also be cited. Here, authors approximate the RTRL with random kronecker factors.
https://papers.nips.cc/paper/7894-approximating-real-time-recurrent-learning-with-random-kronecker-factors
https://www.biorxiv.org/content/10.1101/458570v1

Experiment section: The authors evaluate the proposed method on both synthetic as well as real experiments.

Simulation Problems: The authors use simulation problems to note the robustness of the proposed method to increasing termporal delay in online learning. These tasks show the soundness of the proposed method. Its actually difficult to tell how the proposed method is performing because of the selection of tasks. It might be more interesting to choose same tasks as in UORO paper (https://arxiv.org/abs/1702.05043) and it could be another "baseline" for the proposed method.

Ablations: I liked the fact that the authors consider conducting experiments without state updating, as it could also be  due to using a large buffer rather than explicitly optimizing for the prediction objective.

 Positive: The proposed method could be interesting for learning the state representation for policy gradient RL methods (specifically POMDPs) as the proposed method can leverage use of mini-batches, as well as multiple-updates which is a crucial ingredient to make best use of data collected by the agent interacting with the environment.


**Experience Assessment:**

I have published in this field for several years.

**Review Assessment: Checking Correctness Of Derivations And Theory:**

I assessed the sensibility of the derivations and theory.

**Review Assessment: Checking Correctness Of Experiments:**

I carefully checked the experiments.

**Review Assessment: Thoroughness In Paper Reading:**

I read the paper thoroughly.

---

> ### Author Response · Authors · 2019-11-11
> **Reply to Reviewer 1**
>
> We would like to thank Reviewer 1 for their valuable comments and literature suggestions. We will add discussion of the two referenced papers into our work. [1] provides an interesting approach to obtain an unbiased approximation of the RTRL update, and will be added to the list of other such related approaches. [2] introduces a local, perhaps more biologically plausible variant of RTRL. However, [2] introduces some strong assumptions (like disregarding non-leading order terms, linearity of the RNN) in the analysis of their learning rule, and it is rather unclear if the update rule will lead to a stationary point of any objective.
>
> Simulation Problems: There are potentially two concerns here. The first is the reasons for selecting the tasks, and whether they really test if FPP can perform well. The second is understanding the performance, relative to baselines other than T-BPTT.  We address both below.
>
> Our primary goal in selecting tasks was to allow a controlled experiment where we changed the length of dependencies, to investigate the ability of FPP to capture long-term dependencies in comparison to T-BPTT, on both simpler domains with relatively fewer confounding factors and on realistic datasets. We chose two synthetic problems and two real datasets towards this goal. In CycleWorld, the ability to predict the observation bit is directly linked to how far back the agent is able to remember. StochasticWorld is one level more sophisticated: the agent is required to remember two independent observations from the past which probabilistically influence the present prediction target.
>
> We also chose two real datasets with reasonably long dependencies back in time: Sequential MNIST and character-level prediction of the Penn Treebank. Our character-level prediction task is similar to the character-level prediction task in the UORO paper, but is performed on a real instead of synthetic dataset. We also chose these two because they are common datasets for testing RNNs (see the citations of the Real Datasets subsection in section 5).
>
> For the second potential concern, we agree that UORO would be useful  as a baseline for a completely different approach,  based on RTRL. We will first justify why we did not include initially, and then detail our current efforts in implementing the UORO baseline. We had focused our empirical investigation on T-BPTT for two reasons. First, the computation can be made comparable between T-BPTT and FPP, and the methods are similar in their simplicity. One of our primary goals is to develop simple approaches to train RNNs. Second, T-BPTT remains the standard algorithm for training RNNs, and with sufficiently large T can perform very well. For this reason, we could simply increase T to ensure we reached good performance and then compare for that T and smaller. UORO, and other algorithms that approximate RTRL, are typically more complicated and expensive;they are also relatively new and so not yet as standard.
>
> However, though we thought of it as an apples-to-oranges comparison, UORO would nonetheless provide a baseline comparison to this other class of methods which would absolutely strengthen the results. We are currently implementing the UORO baseline; we will attempt to include it in the revision for the author rebuttal period, and if we cannot finish it in time will include it in a final paper.
>
> [1]: Mujika et al., “Approximating Real-Time Recurrent Learning with Random Kronecker Factors”
> [2]: Murray, “Local online learning in recurrent networks with random feedback.”

---

> > ### Comment · AnonReviewer1 · 2019-11-11
> > **Quick reply.**
> >
> > " [1] provides an interesting approach to obtain an unbiased approximation of the RTRL update, and will be added to the list of other such related approaches. [2] introduces a local, perhaps more biologically plausible variant of RTRL. However, [2] introduces some strong assumptions (like disregarding non-leading order terms, linearity of the RNN) in the analysis of their learning rule, and it is rather unclear if the update rule will lead to a stationary point of any objective. "
> >
> > I agree with this.
> >
> > "We also chose two real datasets with reasonably long dependencies back in time: Sequential MNIST and character-level prediction of the Penn Treebank. "
> >
> > Its hard to actually argue that character level prediction task of PTB actually requires "long-dependencies". Probably a better task would be doing character level prediction for Text8 dataset.
> >
> > "we thought of it as an apples-to-oranges comparison, UORO would nonetheless provide a baseline comparison to this other class of methods which would absolutely strengthen the results."
> >
> > I appreciate. :)

---

> > > ### Author Response · Authors · 2019-11-15
> > > **Preliminary results of UORO**
> > >
> > > We would like to provide some preliminary results of UORO (memory-1 rank-1 UORO). We use the same experimental setting as Figure 3 in our paper. First, we found that UORO is less sample efficient than FPP. The first column of the table below shows the average performance over 5k steps for CycleWorld and 10k steps for StochasticWorld. The second column shows the average performance over 50k steps in both tasks. In CycleWorld, UORO does not learn well during the first 5k steps, while FPP can achieve reasonable accuracy (see Figure 3). We also found that UORO does not perform well in stochastic tasks (e.g. StochasticWorld).
> > >
> > >                                default steps    50k steps
> > > CycleWorld                  9.17%            3.70%
> > > StochasticWorld         0.663             0.662
> > >
> > > We will update the results (including Sequential MNIST and PTB) into the final version of our paper.

---

> > > > ### Comment · AnonReviewer1 · 2019-11-15
> > > > **Thanks.**
> > > >
> > > > Thanks for running those experiments. This definitely helps to improve the paper.

---

> > > > > ### Comment · AnonReviewer1 · 2019-11-15
> > > > > **Sequetial MNIST and PTB**
> > > > >
> > > > > I'm a bit skeptical of the results mentioned about PTB and Sequential MNIST.
> > > > >
> > > > > For PTB as well as sequential MNIST, can authors report their baselines results (i.e when you do full back propagation ?). I just want to make sure, baselines are not "faulty".
> > > > >
> > > > > For reference, authors can see result in Zoneout (https://arxiv.org/abs/1606.01305) or Recurrent Batch Normalization paper (https://arxiv.org/abs/1603.09025). Thanks.

---

### Official Review · AnonReviewer3 · 2019-10-23
**Official Blind Review #3**

**Rating:** 6

**Review:**

The paper proposes an alternative to the truncated back-propagation through time (BPTT) algorithm for training RNNs. An online setting is assumed, which I understand as training RNN as data arrives and not storing too much of the arriving data (although notably the proposed method uses a buffer). The proposed Fixed-Point Propagation algorithm works as follows. It maintains a buffer of the last N RNN states that can be updated. From this buffer at every time step it samples two states that are T steps apart from each other (s_i and s_{i - T}). The RNN is run for T steps starting from s_{i - T}. A loss function is constructed that takes into account the output loss at time i as well as the mismatch between s_i and the new state constructed by running the RNN. The states s_i and s_{i-T}, as well as the RNN parameters are updated based on this loss function.

The novel idea of the paper is therefore a modifiable state buffer for the RNN states. The goal is better computational efficiency than that of T-BPTT.

The paper is mostly clearly written, but I think it is absolutely necessary to move Algorithm 1 to the main text, as well as to add the mini-batch processing (B) and multiple updates (M) to it. This pseudocode was very instrumental for me to understand the algorithm. I confess that I did not read the theory; I don’t think it’s super relevant because in practice convergence to fixed-point will require too many updates.

The empirical comparison with T-BPTT is substantial, but the waters are muddied a bit by imprecise presentation of baselines. For example, when T-BPTT is used for e.g. language modelling, it doesn’t make sense for back-propagate the loss from only the last time step, losses from all time-steps can be back-propagated together. Was this done in T-BPTT and/or FPP? Does T-BPTT use the 100 step buffer somehow? NoOverlap T-BPTT is not explained very well. A very interesting and absolutely necessary baseline is FPP without state updates, but for such a baseline the loss comparing s_t and s_{i-T} should be disabled. Was this done?

In short, the paper must clear show that updating the states in the buffer allows to get same performance with smaller T, compared to the best possible baseline that also uses the buffer but does not update states in it. I am not sure this case is clearly made at the moment.
One further direction authors could explore is that using a very small T but larger B could be more computationally inefficient because parallel computations would be used instead of sequential ones. Besides, from a practical viewpoint and I think it could make sense to also update intermediate states, and not just s_i and s_{i-T}.

Other remarks:
- the legend in Figure 4 is a dashed line, but the curves in the plots are dotted
- the max. cycle length in CycleWorld is not clearly explained in the text, the name CycleWorld is not properly introduced


**Experience Assessment:**

I have read many papers in this area.

**Review Assessment: Checking Correctness Of Derivations And Theory:**

I did not assess the derivations or theory.

**Review Assessment: Checking Correctness Of Experiments:**

I carefully checked the experiments.

**Review Assessment: Thoroughness In Paper Reading:**

I read the paper at least twice and used my best judgement in assessing the paper.

---

> ### Author Response · Authors · 2019-11-11
> **Reply to Reviewer 3**
>
> We thank Reviewer 3 for pointing out the errata and clarity issues; we will make the necessary corrections. We will update the PDF by November 12. We will add the pseudocode into section 3 (with mini-batch processing), we will make the legend in figure 4 consistent, we will improve the explanation of non-overlapping T-BPTT in section 5, and we will explain the CycleWorld environment in section 5.
>
> The main concern of Reviewer 3 was about baselines. We would like to clarify that our problem setting is online prediction, where data constantly arrives in a stream, rather than offline training from a fixed batch of data. The T-BPTT and no-overlap T-BPTT algorithms are adapted to our online problem setting: where only the loss from the last time step are back-propagated, which is standard for training RNNs online. Even though language modeling can often be done offline, it nonetheless serves as a useful realistic task to evaluate FPP and other algorithms under online training.
>
> > ”A very interesting and absolutely necessary baseline is FPP without state updates, but for such a baseline the loss comparing s_t and s_{i-T} should be disabled. Was this done?”
> Yes, this was done. “FPP without State Updates” does not include the quadratic penalty term that compares \hat{s}_i and s_{i - T}.
>
> >”the paper must clear show that updating the states in the buffer allows to get same performance with smaller T, compared to the best possible baseline that also uses the buffer but does not update states in it.”
> We agree. We do believe the comparison to FPP w/o state updating provides this role. Potentially the baseline Reviewer 3 feels is missing is running T-BPTT on the buffer, as if it was a fixed dataset (i.e., offline). In fact, FPP w/o state updating is like using T-BPTT: T steps of back-prop-through time are computed, without the state-loss and by starting from a given state (in this case, whatever the state was at that time). We in fact tried a few other strategies of starting from random states rather than stored states, or periodically updating states in the buffer so that start states were less arbitrary; these choices did not improve performance. It is of course possible that another approach using a buffer, that does not update (or even store) state variables, could be developed that outperforms FPP. However, such an approach is not obvious and would be a novel algorithm. We cannot and do not claim to have definitely demonstrated that one must maintain and update state variables. We do nevertheless claim that the baseline of FPP w/o state updating provides evidence of the importance of state updating.
>
> There are a couple of other comments about future directions, which we appreciate! One of our goals is to take advantage of parallelization with FPP, and so Reviewer 3’s point about increasing B is well-taken. Reviewer 3 also raises an interesting point about updating intermediate states. This is equivalent to setting T=1 and updating consecutive transitions in one mini-batch. There are multiple ways to sample transitions (e.g. sampling consecutive transitions, prioritized sampling or uniform sampling) and perform updates (e.g. choice of T, B and M), given the sound approach for training an RNN with a buffer. These are promising strategies to try, but are additional after first understanding the basic idea; we therefore picked the simplest strategies to show in this paper. A next step is to further investigate how much we can improve performance by more effective sampling approaches, and by further investigating the effects of T, B, and M.

---

> > ### Author Response · Authors · 2019-11-12
> > **Updated pdf**
> >
> > The pdf is updated now with the changes we mentioned in the comment!

---

> > > ### Comment · AnonReviewer3 · 2019-11-15
> > > **Further comments**
> > >
> > > I would like to thank authors for their informative response, as well as for improving presentation in the paper. Some of my concerns regarding the baselines, however, remain.
> > >
> > > Quoting the authors' response:
> > >
> > > > In fact, FPP w/o state updating is like using T-BPTT: T steps of back-prop-through time are computed, without the state-loss and by starting from a given state (in this case, whatever the state was at that time).
> > >
> > > In this is the case, I don't understand by FPP w/o state update is not considered as the main baseline, especially given that, quoting the paper, it is similar to a published "Stored State T-BPTT" method. The comparison between FPP and FPP w/o state update take place in separate figure and only using toy task. It is therefore very hard to understand which percentage of the improvement that FPP brings upon T-BPTT comes just from using the buffer, and not from updating the states in it. Besides, in Figure 6 cross-entropies for StochasticWorld are much higher than those in Figure 3(b). What's different between these experiments?

---

### Official Review · AnonReviewer2 · 2019-10-28
**Official Blind Review #2**

**Rating:** 3

**Review:**

In this paper, the authors reformulate the RNN training objective to explicitly learn the state vectors, and propose an algorithm called Fixed Point Propagation (FPP Algorithm 1). The authors motivate the derivation of FPP in Section 3, provide some theoretical convergence results in Section 4, and demonstrate experiment results in Section 5.

In general, this paper is interesting and well written. The experiment results in Section 5 seem to be very strong. However, I am not familiar with the relevant literature, thus, I am not sure if the authors have compared with the strongest baselines in this field.

I think the paper suffers from the following limitations:

1) Theorem 1 shows that the FPP algorithm converges to a stationary point. However, this result seems to be too weak. Can we say something about the stationary point? Is it a local minimum under some conditions?

2) In the experiments, the authors choose \lambda=1. My understanding is that \lambda is a key meta-parameter of FPP. Please do a better job in justifying this choice.

**Experience Assessment:**

I do not know much about this area.

**Review Assessment: Checking Correctness Of Derivations And Theory:**

I assessed the sensibility of the derivations and theory.

**Review Assessment: Checking Correctness Of Experiments:**

I assessed the sensibility of the experiments.

**Review Assessment: Thoroughness In Paper Reading:**

I read the paper at least twice and used my best judgement in assessing the paper.

---

> ### Author Response · Authors · 2019-11-11
> **Reply to Reviewer 2**
>
> We would like to thank Reviewer 2 for the comments and questions. The biggest concerns seem to be about (1) the strength of the theoretical results, (2) the choice of the key hyperparameter of FPP, and (3) comparing with the strongest baselines in the field.
>
> (1): It is in general difficult to make claims about the quality of an arbitrary stationary point; this difficulty also applies to the vanilla RNN objective. We do partially characterize the stationary points of the FPP objective in terms of recovering stationary points of the RNN objective, in Theorem 2. There are also some theoretical and empirical results on the ability of SGD to converge to local minima (for example, [1, 2]), and avoid getting stuck in saddle points. Because we use SGD, we can rely on such results to suggest we may similarly converge to local minima. The current theory, though, focuses on the first key claims: that the algorithm converges to stationary points (which is non-trivial and is not obviously true of the T-BPTT algorithm) and that there is a connection between the stationary points of FPP and the original RNN objective.
>
> (2): In the Appendix, we plot the sensitivity curve for lambda (Fig. 6). This figure shows that FPP is not particularly sensitive to lambda, though it does also indicate that we could have tuned lambda and gotten even better performance. We opted to show performance in the main body for this default value of lambda = 1 across all experiments, which we actually chose before even seeing these sensitivity plots. In general, the performance of FPP was insensitive to lambda across all our experiments. This is one of the benefits of FPP: we do not require extensive hyperparameter tuning is and yet we achieve consistent stability benefits compared with T-BPTT. We will refer more explicitly to these sensitivity curves in the main body to clarify this point.
>
> (3): We do not claim to introduce a model that outperforms the state-of-the-art on any particular dataset (e.g. on GLUE). SOTA performance requires SOTA algorithms, SOTA meta-parameter tuning,  SOTA implementations, and at times specialized hardware. This paper is about a new algorithm for training RNNs. Based on our results we expect many high-performance systems could be built on top of FPP (perhaps even SOTA on some of these data-sets). But this is well beyond the scope of this paper. Think of this paper as introducing a new algorithm, neither a complete learning system nor a SOTA claim. Certainly there is room for all three types of papers in ICLR as their contributions are very different. Our main contribution was to introduce a novel approach for RNN training and show that it is more robust than T-BPTT. Future work in adapting FPP to more modern RNN architectures would be interesting, but is not in the scope of this paper.
>
> [1] Choromanska, et al., “The Loss Surfaces of Multilayer Networks”
> [2] Lee et al., “Gradient Descent Only Converges to Minimizers”

---

### Decision · Program_Chairs · 2019-12-19

**Decision:**

Accept (Poster)

**Comment:**

The paper proposes an alternative to BPTT for training recurrent neural networks based on an explicit state variable, which is trained to improve both the prediction accuracy and the prediction of the next state. One of the benefits of the methods is that it can be used for online training, where BPTT cannot be used in its exact form. Theoretical analysis is developed to show that the algorithm converges to a fixed point. Overall, the reviewers appreciate the clarity of the paper, and find the theory and the experimental evaluation to be reasonably well balanced. After a round of discussion, the authors improved the paper according to the reviews. The final assessments are overall positive, and I’m therefore recommending accepting this paper.